# LogicBench: Towards Systematic Evaluation of Logical Reasoning Ability of Large Language Models

## Abstract

Recently developed large language models (LLMs) have been shown to perform remarkably well on a wide range of language understanding tasks. But, can they really "reason" over the natural language? This question has been receiving significant research attention and a number of reasoning skills such as commonsense, numerical, and qualitative have been studied. However, the crucial skill pertaining to 'logical reasoning' has remained underexplored. Existing work investigating this reasoning ability has focused only on a couple of inference rules (such as modus ponens and modus tollens) of propositional and first-order logic. To enable systematic evaluation of logical reasoning, we introduce *LogicBench*, a natural language question-answering dataset encompassing 25 different reasoning patterns spanning over propositional, first-order, and non-monotonic logics. Key steps of our dataset construction consist of (1) controlled generation of sentences and their negations containing different ontologies, (2) *(context, question, answer)* triplets creation using heuristically designed templates, and (3) semantic variations of triplets adding more diversity. We present a comprehensive evaluation with a range of LLMs such as GPT-4, GPT-3, ChatGPT, and FLAN-T5 using chain-of-thought prompting in both zero-shot and few-shot settings. Experimental results show that existing LLMs do not fare well on *LogicBench*; especially, they struggle on instances involving complex reasoning and negations. Furthermore, they sometimes tend to prioritize parametric knowledge over contextual information and overlook the correct logical reasoning chain. In addition, we also show that LLMs trained using our data exhibit a better understanding of logical reasoning leading to performance improvements on several existing logical reasoning datasets such as LogicNLI, FOLIO, LogiQA, and ReClor.[1]

## 1 Introduction

Large language models such as GPT-3 (Brown et al., 2020b), ChatGPT, and FLAN (Wei et al., 2021) have made remarkable progress in NLP research enabling machines to perform a variety of language tasks that were previously thought to be exclusive to humans (OpenAI, 2023; Brown et al., 2020a; Zhao et al., 2023). However, the ability of these LLMs to reason "logically" over natural language text remains under-explored, even though logical reasoning is a fundamental aspect of intelligence and a crucial requirement for many practical applications, such as question-answering systems (Khashabi, 2019) and conversational agents (Beygi et al., 2022). Although several datasets have been proposed (Clark et al., 2021; Tian et al., 2021; Joshi et al., 2020; Saeed et al., 2021) to evaluate the logical reasoning capabilities of LLMs, these datasets are limited in their scope by (1) not evaluating logical reasoning independently of other forms of reasoning such as LogiQA (Liu et al., 2021a) and ReClor (Yu et al., 2020); and (2) evaluating only a single type of logic and covering only few logical inference rules as done in FOLIO (Han et al., 2022) and ProntoQA (Saparov & He, 2023). Thus, our aim in this work is to address the lacuna of having more comprehensive evaluation dataset for LLMs.

To this end, we propose *LogicBench*, a systematically created question-answering dataset for the evaluation of logical reasoning ability. As illustrated in Figure 1, *LogicBench* includes a total of

---

[1]Data is available at `https://anonymous.4open.science/r/LogicBench-EEBB`

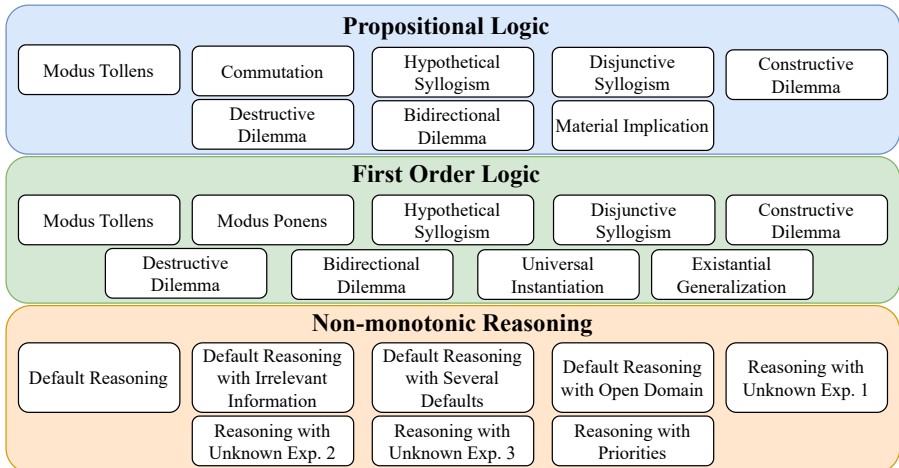

Figure 1: Comprehensive representation of different inference rules and reasoning patterns covered by propositional, first-order, and non-monotonic logics. *Exp.* indicates Expectation

25 reasoning patterns across 'propositional, first-order, and non-monotonic' logics. To evaluate LLMs, we formulate a binary classification task in *LogicBench* in which the context represents logical statements and the models have to determine whether a conclusion given in the question is logically entailed by the context. Examples instances of various reasoning patterns are presented in Table 4 and App. C. To construct *LogicBench*, we use a three-stage procedure (refer to §3). In the first stage, we prompt GPT-3 to generate a variety of coherent natural language sentences consisting of different 'ontologies' (i.e., a collection of concepts such as car, person, and animals) and their corresponding negations (refer to §3.2.1). Then, in the second stage, we generate *(context, question, answer)* triplets using heuristically designed templates based on the inference rules and patterns. Finally, in the third stage, we generate semantics preserving and inverting variations of these logical rules by incorporating negations.

We evaluate a range of LLMs on *LogicBench* including GPT-4, GPT-3 (Brown et al., 2020b), ChatGPT, FLAN-T5 (Wei et al., 2021), and Tk-instruct (Wang et al., 2022b) using chain-of-thought prompting (Wei et al., 2022). In particular, we measure the accuracy of LLMs predictions on the binary classification task. Our experiments result in several interesting findings such as LLMs often struggle to reason over complex logical contexts and encounter difficulties with inference rules involving negations. Experimental results reveal that these models struggle with respect to many of the inference rules and patterns, suggesting significant room for improvement in their logical reasoning abilities. In addition, we synthetically augment *LogicBench* and train T5-large. Our initial experimental results show that this improves the logical reasoning ability of existing models leading to performance improvement on other logic datasets, LogicNLI, and FOLIO ($\sim 2\%$ on an average), and shows competitive performance on LogiQA and ReClor. In summary, our contributions are:

1. Introducing *LogicBench*, a systematically created dataset to assess the logical reasoning capabilities of LLMs across propositional, first-order, and non-monotonic logics. This benchmark will be publicly available for evaluation and training purposes.
2. We propose a three-stage method to construct *LogicBench* consisting of GPT-3 to generate coherent natural language sentences using prompts and a template-based module to convert them into logical rules. By assessing the performance of existing LLMs, we gain insights into their logical reasoning abilities which further leads to several interesting findings.
3. To the best of the authors' knowledge, this is the first benchmark to study non-monotonic reasoning, as well as various inference rules in propositional and first-order logics including hypothetical and disjunctive syllogism; and bidirectional, constructive, and destructive dilemmas in NLP domain.

## 2 RELATED WORK

As LLMs continue to evolve rapidly, it becomes increasingly crucial to evaluate their diverse reasoning capabilities, as well as those of forthcoming LLMs. Past attempts have been made to evaluate the

logical reasoning abilities of these models. LogiQA (Liu et al., 2021a) and ReClor (Yu et al., 2020) have made notable contributions by compiling multichoice questions from standardized examinations that demand diverse forms of logical reasoning. However, in contrast to our *LogicBench*, these datasets involve mixed forms of reasoning and do not focus on assessing logical reasoning in isolation.

A few past attempts have been made to create datasets to evaluate only logical reasoning while excluding other forms of reasoning. For example, CLUTTER (Sinha et al., 2019) covers inductive reasoning, Hahn et al. (2021) covers temporal logic, and Ruletaker (Clark et al., 2021) evaluates whether a transformer-based model

| Dataset | Logic Covered | | | Inference Rules/Axioms Provided with Data | Generation Code Available |
|---------|-----|-----|-----|------|------|
| | PL | FL | NM | | |
| Ruletaker | ✗ | ✓ | ✗ | ✗ | Human-annotated |
| LogicNLI | ✗ | ✓ | ✗ | ✗ | Semi-automated |
| ProofWriter | ✓ | ✓ | ✗ | ✗ | ✗ |
| FOLIO | ✗ | ✓ | ✗ | ✗ | Human-annotated |
| SimpleLogic | ✓ | ✗ | ✗ | ✗ | ✓ |
| ProntoQA | ✗ | ✓ | ✗ | ✓ | ✓ |
| **LogicBench** | ✓ | ✓ | ✓ | ✓ | ✓ |

Table 1: Comparison of *LogicBench* with existing datasets

emulates deductive reasoning over synthetically generated statements in a limited setting. LogicNLI (Tian et al., 2021) introduced a diagnostic benchmark for FOL reasoning, with the dataset constructed by automatically generating logic expressions and replacing the entity and attribute placeholders.

Our proposed dataset is similar (in terms of task formulation) to ProofWriter (Tafjord et al., 2021), FOLIO (Han et al., 2022), and ProntoQA (Saparov & He, 2023) which are QA datasets designed to test reasoning ability. ProofWriter provides multi-hop proofs for each example, while FOLIO gives diverse and complex logical expressions, however, it is only limited to FOL. ProntoQA (Saparov & He, 2023) provides explanation and reasoning steps but is limited to modus ponens in FOL. Nevertheless, several crucial attributes motivated us to create *LogicBench* (see Table 1 for comparison). Additional datasets for evaluating logical reasoning also exist such as SimpleLogic (Zhang et al., 2022) provides a class of logical reasoning problems, TaxiNLI (Joshi et al., 2020) introduces logical taxonomy in NLI task and RuleBert (Saeed et al., 2021) covers only soft logical rules. In summary, *LogicBench* evaluates logical reasoning in isolation and provides diverse inference rules and logic types compared to existing datasets. Extended related work is discussed in App. B.

# 3 LOGICBENCH

## 3.1 LOGICS TYPES

**Propositional Logic (PL)** Propositional logic employs a collection of statements or propositions (denoted as $\mathcal{P} = p_1, p_2, ..., p_n$, where $p_i$ represents a proposition) and builds upon them using logical connectives such as '∧', '∨', '→', '↔', and '¬'. Several inference rules for propositional logic have been defined using which given a set of premises, one can derive a sound conclusion. To illustrate this, let us consider two propositions: $p_1$, which states "It is raining," and $p_2$, which states "It is cloudy." From these propositions, we can construct a context/knowledge base (KB) consisting of two premises: (1) $p_1 \rightarrow p_2$ and (2) $p_1$. Based on this KB, we can conclude $p_2$. This inference rule is written as $((p_1 \rightarrow p_2) \wedge p_1) \vdash p_2$ and is known as 'Modus Ponens'. In our study, we explore nine distinct inference rules of propositional logic, extensions of seven of them with one-variable and a universal quantifier, and two axioms of first-order logic as shown in Table 2. These inference rules provide a systematic framework for deriving valid conclusions.

**First-order Logic (FOL)** In this work, we consider a restricted set of logical axioms for FOL that utilize quantifiers, ∀ (universal quantifier) and ∃ (existential quantifier). The universal quantifier (∀) denotes that a statement holds true for all instances within a specific category. In contrast, the existential quantifier (∃) indicates that a statement is true for at least one instance within its scope. For instance, a simple extension of propositional 'Modus Ponens' is an inference rule where given the premises $\forall(p(x) \rightarrow q(x))$ and $p(a)$, we conclude $q(a)$ (e.g., given "All kings are greedy" and "Sam is a king", we can conclude "Sam is greedy"). Here, we explore two axioms (EG and UI - details are presented in App. C.3) and various inference rules that incorporate the quantifiers (shown in Table 2).

| Names | Propositional Logic | Extension to a (restricted) First-order Logic |
|---|---|---|
| MP | $((p \to q) \land p) \vdash q$ | $(\forall x(p(x) \to q(x)) \land p(a)) \vdash q(a)$ |
| MT | $((p \to q) \land \neg q) \vdash \neg p$ | $(\forall x(p(x) \to q(x)) \land \neg q(a)) \vdash \neg p(a)$ |
| HS | $((p \to q)) \land (q \to r)) \vdash (p \to r)$ | $(\forall x((p(x) \to q(x)) \land (q(x) \to r(x))) \vdash (p(a) \to r(a))$ |
| DS | $((p \lor q) \land \neg p) \vdash q$ | $(\forall x(p(x) \lor q(x)) \land \neg p(a)) \vdash q(a)$ |
| CD | $((p \to q) \land (r \to s) \land (p \lor r)) \vdash (q \lor s)$ | $(\forall x((p(x) \to q(x)) \land (r(x) \to s(x))) \land (p(a) \lor r(a))) \vdash (q(a) \lor s(a))$ |
| DD | $((p \to q) \land (r \to s) \land (\neg q \lor \neg s)) \vdash (\neg p \lor \neg r)$ | $(\forall x((p(x) \to q(x)) \land (r(x) \to s(x))) \land (\neg q(a) \lor \neg s(a))) \vdash (\neg p(a) \lor \neg r(a))$ |
| BD | $((p \to q) \land (r \to s) \land (p \lor \neg s)) \vdash (q \lor \neg r)$ | $(\forall x((p(x) \to q(x)) \land (r(x) \to s(x))) \land (p(a) \lor \neg s(a))) \vdash (q(a) \lor \neg r(a))$ |
| CT | $(p \lor q) \vdash (q \lor p)$ | - |
| MI | $(p \to q) \vdash (\neg p \lor q)$ | - |
| EG | - | $P(a) \Rightarrow \exists x P(x)$ |
| UI | - | $\forall x A \Rightarrow A\{x \mapsto a\}$ |

Table 2: Inference rules and (two) axioms that establish the relationship between premises and conclusions. MP: Modus Ponens, MT: Modus Tollens, HS: Hypothetical Syllogism, DS: Disjunctive Syllogism, CD: Constructive Dilemma, DD: Destructive Dilemma, BD: Bidirectional Dilemma, CT: Commutation, MI: Material Implication, EG: Existential Generalization, UI: Universal Instantiation

**Non-monotonic (NM) Reasoning**    In this work, we analyze a range of logical reasoning templates in NM logics involving "Default Reasoning," "Reasoning about Unknown Expectations," and "Reasoning about Priorities." These templates are inspired by the compilation (Lifschitz, 1989) made in 1989 to evaluate the abilities of various non-monotonic logics that were being developed at that time. Below Table 3 shows examples of NM reasoning. Additional examples are given in App. C.4.

| Basic Default Reasoning | Default Reasoning with Irrelevant Information |
|---|---|
| Context: Blocks A and B are heavy. Heavy blocks are typically located on the table. A is not on the table. 

 Conclusion: B is on the table. | Context: Blocks A and B are heavy. Heavy blocks are typically located on the table. A is not on the table. B is red. 

 Conclusion: B is on the table. |
| **Reasoning about Unknown Expectations** | **Reasoning about Priorities** |
| Context: Blocks A, B, and C are heavy. Heavy blocks are normally located on the table. At least one of A, B is not on the table. 

 Conclusion: C is on the table. Exactly one of A, B is not on the table. | Context: Jack asserts that block A is on the table. Mary asserts that block A is not on the table. When people assert something, they are normally right. 

 Conclusion: If Mary's evidence is more reliable than Jack's. then block A is not on the table |

Table 3: Illustrative examples of non-monotonic reasoning adapted from Lifschitz (1989)

A key aspect of NM logics is to formalize notions such as "normally," "typically," and "usually" that are not directly formalizable using classical quantifiers in the first-order setting. The general rule "Heavy blocks are normally located on the table" does not imply that "All heavy blocks are always located on the table". Rather, this rule allows for exceptions. Our work explores various NM reasoning patterns, as depicted in Figure 1, to delve deeper into the nuances of this type of reasoning.

## 3.2    DATA CREATION

Our data creation procedure, illustrated in Figure 2, consists of three stages:

1. **Sentence Generation:** Starting with a given prompt, we generate coherent sentences and their negations that incorporate different ontologies.
2. **NL Conversion:** Using predefined templates of reasoning patterns based on their formal expressions, we convert the generated sentences into *(context, question, answer)* triplets.
3. **Variation Generation:** We generate semantically preserving and inverting variations of these triplets to add more diversity.

By following this method, we construct *LogicBench*, and examples of generated data corresponding to each logic type and reasoning patterns are presented in App. C.

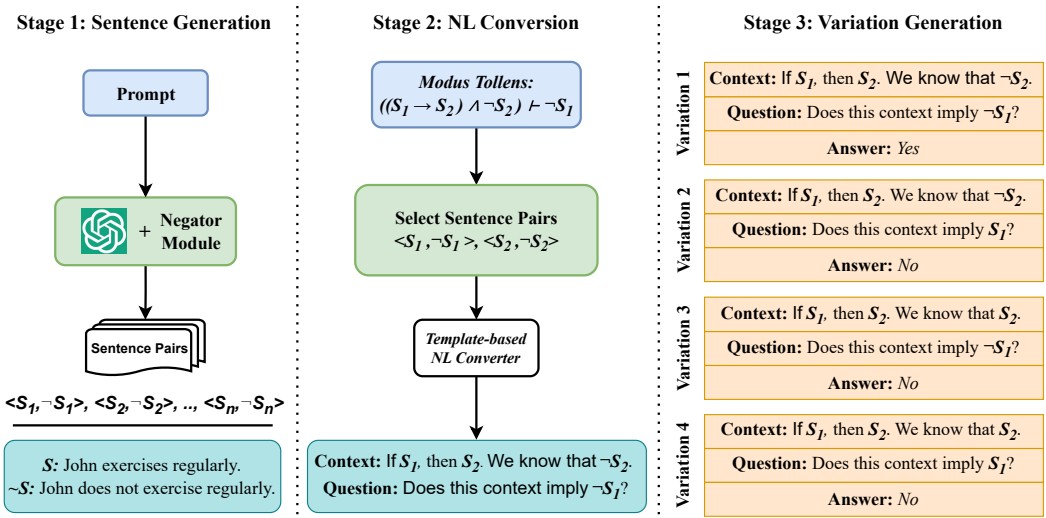

Figure 2: Schematic representation of three-stage procedure for data creation. NL: Natural Language

### 3.2.1 SENTENCE GENERATION

Here, the first step is to generate sentences with diverse *ontologies*. An ontology represents a collection of concepts (e.g. car, person, animals, etc.) along with their corresponding associated properties. To generate these sentences, we prompt the GPT-3 model with instructions tailored for each inference rule. The prompt schema, as depicted in Figure 3, comprise three crucial components:

*Definition* provides a detailed explanation of the task and offers a natural language representation of the reasoning pattern for which we are generating sentences.

*Examples* provide sample sentences that need to be generated. We also illustrate how these sentences will be utilized in later stages, emphasizing the importance of coherence and the inclusion of relevant ontological concepts.

*Format* We provide specific formatting instructions to guide the generation of sentences.

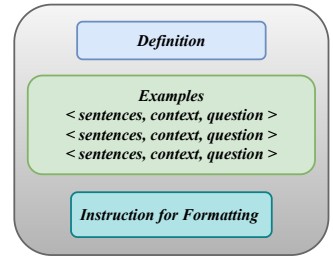

An example of a prompt corresponding to the 'Modus Tollens' from PL is presented in App. A for better illustration. Note that our objective at this stage is not to generate logical sentences but rather to generate a diverse and coherent set of sentences that encompass various concepts. We also create a negation sentence corresponding to each generated sentence[2]. In this work, the scope of generating negations is simple (refer to Appendix C for examples), however, negations can be more complicated in the case of logic. These generated sentences will be combined with logical connectives in a later stage to form context and questions.

Figure 3: Schematic representation of prompt.

### 3.2.2 NL CONVERSION

We leverage the formal expressions of reasoning patterns to create templates that establish the desired NL formulation for each logical connective. For instance, implication: "$p \rightarrow q$" is expressed as "If $p$, then $q$", conjunction: "$p \wedge q$" as "$p$ and $q$.", and disjunction: "$p \vee q$" as "At least one of the following is true: (1) $p$ and (2) $q$. Note that we do not know which of (1) and (2) is true. It is possible that only (1) is true, or only (2) is true, or both are true." since understanding the logical implication of 'or' when integrated into logical formulations posed challenges to both humans and models.

---

[2]We use `https://github.com/dmlls/negate` to generate negated sentences

With these established formulations, we proceed to utilize the sentences generated in §3.2.1 to create the context and questions corresponding to reasoning patterns. For instance, let's consider the "Modus Tollens" from PL $(((p \rightarrow q) \wedge \neg q) \vdash \neg p)$, and the "Bidirectional Dilemma" from FOL $(\forall x((p(x) \rightarrow q(x)) \wedge (r(x) \rightarrow s(x))) \wedge (p(a) \vee \neg s(a))) \vdash (q(a) \vee \neg r(a)))$. Table 4 presents examples of logical context and questions for these inference rules and App. D showcases further examples corresponding to each inference rule and patterns from *LogicBench*.

| Inference Rule | Generated Sentences in Stage 1 | Context and Question |
|---|---|---|
| Modus Tollens | p: Liam finished his work early. 
 ¬p: Liam did not finish his work early. 
 q: he will order pizza for dinner. 
 ¬q: he will not order pizza for dinner. | **Context:** If Liam finishes his work early, then he will order pizza for dinner. 

 **Question:** If he won't order pizza for dinner, does this imply that Liam didn't finish his work early? |
| Bidirectional Dilemma | p(x): someone drinks lots of water 
 q(x): they will feel hydrated 
 r(x): they eat too much sugar 
 s(x): they will experience a sugar crash 
 p(a): Jane drinks lots of water 
 ¬p(a): Jane does not drink lots of water 
 q(a): she will feel hydrated 
 ¬q(a): she will not feel hydrated 
 r(a): she eats too much sugar 
 ¬r(a): she does not eat too much sugar 
 s(a): she will experience a sugar crash 
 ¬s(a): she will not experience a sugar crash | **Context:** If someone drinks lots of water, then they will feel hydrated. If they eat too much sugar, then they will experience a sugar crash. We know that at least one of the following is true (1) Jane drinks lots of water and (2) she won't experience a sugar crash. Note that we do not know which ones of (1) and (2) are true. It might be the case that only (1) is true, or only (2) is true or both are true. 

 **Question:** If at least one of (1) and (2) is true, can we say, at least one of the following must always be true? (a) she will feel hydrated and (b) she doesn't eat too much sugar. |

Table 4: Illustrative examples of logical context and questions created using sentences that are generated in the first stage §3.2.1.

### 3.2.3 VARIATION GENERATION

After generating the context and questions in §3.2.2, we generate semantically preserving and inverting variations of questions. Let's consider the example of "Modus Tollens" from Table 4, where the question is: "If he won't order pizza for dinner, does this imply that Liam didn't finish his work early?" In this question, we observe two propositions: $s_1$, representing the statement "Liam didn't finish his work early," and $s_2$, representing the statement "He won't order pizza for dinner." By perturbing these propositions, we can create four possible tuples: $< s_1, s_2 >, < \neg s_1, s_2 >, < s_1, \neg s_2 >, < \neg s_1, \neg s_2 >$. Each tuple represents a combination of true or negation values for the propositions. Although it is possible to create more combinations from $< s_1, \neg s_1 >$, and $< s_2, \neg s_2 >$, we refine and restrict the set of triplets to exclude those that undermine the validity of the inference rule. Moreover, we do not generate variations for the context since it offers no substantial diversity in the dataset. To generate question variations, we replace the propositions in the original question with the corresponding tuples from the generated variations, hence, adding more diversity to *LogicBench*. This process allows us to create different variations of the question, as illustrated in Figure 2 (Step 3). More examples of question variations are in App. C.

### 3.3 STATISTICS AND QUALITATIVE ANALYSIS

**Statistics** We introduce two versions of our proposed dataset: *LogicBench(Eval)* and *LogicBench(Aug)*. Statistics of both versions are presented in Table 5. Here, *LogicBench(Eval)* is created using the above

| Dataset | # of Instances per Axiom | Total # of Instances | Total # of Instances (Including Variations) |
|---|---|---|---|
| *LogicBench(Eval)* | 20 | 500 | 1720 |
| *LogicBench(Aug)* | 150 | 3750 | 12908 |

Table 5: Statistics of the *LogicBench(Eval)* and *LogicBench(Aug)*

method along with human-in-loop to ensure the quality of generated data, whereas *LogicBench(Aug)* is only a synthetically augmented version for training purposes. Here, we use "human-in-loop" for the authors who conducted qualitative analysis of data. For LogicBench(Eval), out of 1720, 531 samples are for 'yes' and 1189 samples are for 'no'. LogicBench(Aug) follows the same ratio.

**Quality of Data**   Throughout the data generation phase of *LogicBench(Eval)*, the authors conduct a review of the logical formations to ensure they follow the intended logical structure. We examine each reasoning pattern for any potential discrepancies, ensuring that they were logically sound and correctly represented the intended relationships between propositions. To further support the integrity and reliability of the benchmark, we performed a small-scale human study presented in Appendix L. In addition to the logical formation, we also dedicated considerable effort to eliminating typos and validating the grammar. Furthermore, we also analyze the diversity in terms of different ontology and the logical nature of the *LogicBench(Eval)* (presented in App. C.1). We believe that these two versions aim to accommodate different evaluation and training needs to explore logical reasoning.

## 4   RESULTS AND ANALYSIS

### 4.1   EXPERIMENTAL SETUP

**Task Formulation**   We formulate binary classification task using *LogicBench* to evaluate the logical reasoning ability of LLMs. Let us consider a set of data instances $\mathcal{I}_{a,L}$ corresponding to axiom $a$ and logic type $L$. In this set, $i^{th}$ instance is represented as $\mathcal{I}_{a,L}^{i} = \{(c_i, Q_i)\}$ where $c_i$ represents context and $Q_i = \{q_1, q_2, ..., q_n\}$ represents set of question and its variations corresponding to $i^{th}$ instance. As discussed in §3, each context ($c$) represents logical rules (e.g., All cats have fur. Tom is a cat.) and question ($q$) represents the conclusion (e.g., Does Tom have fur?). To each context and question pair, i.e., $< c, q >$, we assign a label from the set $\mathcal{Y} = \{Yes, No\}$. We assign a label $Yes$ if the conclusion logically entails the context, otherwise, assign a label $No$. To evaluate any LLMs on this setup, we provide $< p, c, q >$ as input to predict a label from $\mathcal{Y}$ where $p$ is a natural language prompt. In this work, we use chain-of-thought prompts to evaluate LLMs.

**Experiments**   We evaluate a range of prompting models (i.e., GPT-4, GPT-3 (davinci-003) and ChatGPT), and instruction-tuned models (FLAN-T5 (3B) and Tk-instruct (3B)) on *LogicBench(Eval)*. Each model is evaluated in a zero-shot setting where the chain-of-thought prompt is provided to the model without any in-context examples. This approach allows us to determine LLM's inherent ability to do logical reasoning (based on pre-training), as we can not expect that various logical inference rules/patterns will always be made part of prompts. However, we do evaluate these models in a few-shot setting, and present the results in App. E. We also evaluate these models with and without chain-of-thought prompting, presented in App. F. In addition, we present exploratory – only exploratory because of the limited availability of inference APIs – analysis over Bard in App. G.

In addition, we trained the T5-large model on the *LogicBench(Aug)* resulting in a model named LogicT5. Furthermore, we performed fine-tuning on four other logical reasoning datasets: LogiQA, Reclor, LogicNLI, and FOLIO. Our experiments were carried out in two settings: single-task (fine-tuning and evaluation on one dataset) and multi-task (fine-tuning on all four datasets combined, with separate evaluations for each dataset). App. H describes a detailed experimental setup.

**Metrics**   Here, we evaluate performance in terms of accuracy corresponding to each label, i.e., $A(Yes)$ and $A(No)$. We evaluate each model on three different chain-of-thought prompts and report average results across these prompts. All prompts used for experiments are described in App. H.

### 4.2   BENCHMARK RESULTS

Table 6 represents inference rule-wise performance, and label-wise accuracy ($A(Yes)$ and $A(No)$) corresponding to each LLMs. Here, we focus on analyzing the $A(Yes)$ since the aim is to understand the model's logical reasoning capabilities in answering the question where the conclusion entails the logical context. Table 6 provides valuable insights into the performance of different models on various logic types. For PL, instruction-tuned models FLAN-T5, and Tk-instruct achieve 41.71%, and 30.11% $A(Yes)$, respectively. GPT-3 demonstrates a performance of 39.65%, ChatGPT achieves 46.16%, and GPT-4 shows performance of 44.25%. This indicates the challenge of classical logical reasoning even for larger LLMs like ChatGPT and GPT-4. Moving on to FOL, these models showcase performance accuracy of 64.34%, 60.25%, 52.27%, 66.78%, and 59.53% (on average) for FLAN-T5, Tk-instruct, GPT-3, ChatGPT, and GPT-4, respectively. On the NM reasoning, these models show

| Type | Rules | FLAN-T5 | | Tk-instruct | | GPT-3 | | ChatGPT | | GPT-4 | |
|---|---|---|---|---|---|---|---|---|---|---|---|
| | | $A(No)$ | $A(Yes)$ | $A(No)$ | $A(Yes)$ | $A(No)$ | $A(Yes)$ | $A(No)$ | $A(Yes)$ | $A(No)$ | $A(Yes)$ |
| PL | HS | $95.50_{0.08}$ | $44.42_{0.18}$ | $97.22_{0.01}$ | $91.82_{0.05}$ | $100_{0.00}$ | $50.08_{0.03}$ | $100_{0.00}$ | $70.71_{0.17}$ | $98.76_{0.01}$ | $96.45_{0.03}$ |
| | DS | $70.47_{0.03}$ | $13.33_{0.12}$ | $73.45_{0.01}$ | $0.00_{0.00}$ | $84.42_{0.07}$ | $50.29_{0.07}$ | $77.42_{0.02}$ | $41.27_{0.36}$ | $76.22_{0.02}$ | $26.67_{0.46}$ |
| | CD | $98.72_{0.02}$ | $60.50_{0.09}$ | $75.86_{0.00}$ | $6.67_{0.12}$ | $98.96_{0.02}$ | $38.58_{0.04}$ | $79.43_{0.06}$ | $68.75_{0.31}$ | $76.13_{0.02}$ | $33.33_{0.58}$ |
| | DD | $57.70_{0.26}$ | $6.06_{0.10}$ | $75.55_{0.04}$ | $21.11_{0.02}$ | $88.80_{0.01}$ | $32.28_{0.05}$ | $81.68_{0.06}$ | $29.06_{0.26}$ | $75.93_{0.02}$ | $33.33_{0.58}$ |
| | BD | $76.02_{0.17}$ | $85.82_{0.06}$ | $87.89_{0.06}$ | $55.27_{0.12}$ | $97.77_{0.02}$ | $35.79_{0.04}$ | $78.40_{0.02}$ | $58.40_{0.37}$ | $75.03_{0.003}$ | $20.00_{0.35}$ |
| | MT | $84.72_{0.08}$ | $48.69_{0.36}$ | $70.00_{0.00}$ | $10.00_{0.00}$ | $88.77_{0.02}$ | $40.82_{0.05}$ | $84.59_{0.08}$ | $48.81_{0.19}$ | $94.15_{0.06}$ | $85.19_{0.07}$ |
| | MI | $78.40_{0.04}$ | $0.00_{0.00}$ | $67.03_{0.04}$ | $0.00_{0.00}$ | $94.44_{0.10}$ | $26.74_{0.02}$ | $73.02_{0.03}$ | $14.82_{0.13}$ | $81.32_{0.07}$ | $46.03_{0.18}$ |
| | CT | $84.12_{0.04}$ | $74.81_{0.07}$ | $89.45_{0.04}$ | $56.09_{0.04}$ | $98.29_{0.03}$ | $42.61_{0.04}$ | $75.92_{0.01}$ | $37.50_{0.33}$ | $76.56_{0.03}$ | $12.96_{0.22}$ |
| | Avg | $80.71_{0.12}$ | $41.71_{0.13}$ | $79.18_{0.02}$ | $30.11_{0.04}$ | $93.93_{0.03}$ | $39.65_{0.04}$ | $81.31_{0.03}$ | $46.16_{0.26}$ | $81.76_{0.03}$ | $44.25_{0.31}$ |
| FOL | EG | $92.59_{0.13}$ | $100_{0.00}$ | $95.24_{0.00}$ | $100_{0.00}$ | $90.69_{0.08}$ | $91.61_{0.03}$ | $92.62_{0.07}$ | $100_{0.00}$ | $100_{0.00}$ | $100_{0.00}$ |
| | UI | $90.18_{0.01}$ | $100_{0.00}$ | $90.00_{0.00}$ | $89.82_{0.01}$ | $95.96_{0.04}$ | $79.57_{0.03}$ | $89.11_{0.02}$ | $94.54_{0.002}$ | $98.41_{0.03}$ | $100_{0.00}$ |
| | MP | $81.26_{0.09}$ | $84.47_{0.05}$ | $77.78_{0.00}$ | $100_{0.00}$ | $86.62_{0.03}$ | $58.13_{0.02}$ | $82.32_{0.04}$ | $75.62_{0.07}$ | $81.56_{0.002}$ | $98.41_{0.03}$ |
| | HS | $96.46_{0.04}$ | $49.67_{0.05}$ | $98.25_{0.00}$ | $82.61_{0.00}$ | $100_{0.00}$ | $48.01_{0.01}$ | $97.78_{0.04}$ | $65.88_{0.12}$ | $97.93_{0.02}$ | $91.49_{0.07}$ |
| | DS | $82.45_{0.09}$ | $16.67_{0.29}$ | $74.36_{0.00}$ | $0.00_{0.00}$ | $80.43_{0.01}$ | $58.25_{0.11}$ | $81.14_{0.04}$ | $83.97_{0.14}$ | $75.32_{0.01}$ | $33.33_{0.58}$ |
| | CD | $90.84_{0.03}$ | $67.78_{0.07}$ | $79.00_{0.00}$ | $78.79_{0.37}$ | $100_{0.00}$ | $38.73_{0.06}$ | $79.34_{0.02}$ | $44.97_{0.40}$ | $75.46_{0.08}$ | $33.33_{0.58}$ |
| | DD | $72.02_{0.19}$ | $37.14_{0.03}$ | $72.14_{0.00}$ | $13.98_{0.05}$ | $75.58_{0.07}$ | $26.16_{0.04}$ | $75.60_{0.01}$ | $19.44_{0.17}$ | $76.56_{0.03}$ | $14.29_{0.25}$ |
| | BD | $88.57_{0.10}$ | $66.71_{0.09}$ | $96.93_{0.02}$ | $75.10_{0.27}$ | $92.30_{0.03}$ | $32.03_{0.04}$ | $80.43_{0.05}$ | $50.00_{0.44}$ | $78.30_{0.06}$ | $0.00_{0.00}$ |
| | MT | $87.61_{0.02}$ | $56.67_{0.12}$ | $69.11_{0.01}$ | $1.96_{0.03}$ | $84.05_{0.04}$ | $37.97_{0.04}$ | $90.13_{0.03}$ | $66.61_{0.09}$ | $93.76_{0.03}$ | $64.89_{0.04}$ |
| | Avg | $86.89_{0.08}$ | $64.34_{0.08}$ | $83.64_{0.003}$ | $60.25_{0.08}$ | $89.52_{0.03}$ | $52.27_{0.04}$ | $85.39_{0.04}$ | $66.78_{0.16}$ | $86.37_{0.02}$ | $59.53_{0.17}$ |
| NM | DRI | $67.35_{0.02}$ | $79.61_{0.03}$ | $52.63_{0.00}$ | $100_{0.00}$ | $62.74_{0.02}$ | $64.27_{0.06}$ | $58.37_{0.05}$ | $100_{0.00}$ | $71.43_{0.00}$ | $100_{0.00}$ |
| | DRS | $72.96_{0.04}$ | $0.00_{0.00}$ | $72.97_{0.00}$ | $0.00_{0.00}$ | $64.89_{0.01}$ | $0.00_{0.00}$ | $70.40_{0.01}$ | $2.08_{0.04}$ | $68.52_{0.02}$ | $13.21_{0.03}$ |
| | DRD | $73.18_{0.07}$ | $95.54_{0.04}$ | $64.52_{0.00}$ | $100_{0.00}$ | $85.76_{0.03}$ | $74.92_{0.05}$ | $56.26_{0.04}$ | $100_{0.00}$ | $98.41_{0.03}$ | $100_{0.00}$ |
| | DRO | $55.21_{0.02}$ | $86.11_{0.13}$ | $51.28_{0.00}$ | $100_{0.00}$ | $69.86_{0.04}$ | $64.77_{0.06}$ | $51.73_{0.04}$ | $100_{0.00}$ | $50.85_{0.01}$ | $66.67_{0.58}$ |
| | RE1 | $75.05_{0.04}$ | $26.80_{0.04}$ | $75.00_{0.00}$ | $0.00_{0.00}$ | $84.97_{0.08}$ | $35.14_{0.03}$ | $78.54_{0.05}$ | $28.23_{0.12}$ | $75.85_{0.01}$ | $50.00_{0.50}$ |
| | RE2 | $74.82_{0.14}$ | $63.89_{0.55}$ | $75.00_{0.00}$ | $0.00_{0.00}$ | $48.70_{0.01}$ | $0.00_{0.00}$ | $49.57_{0.01}$ | $16.67_{0.29}$ | $50.00_{0.50}$ | $0.00_{0.00}$ |
| | RE3 | $50.05_{0.03}$ | $56.65_{0.10}$ | $57.24_{0.04}$ | $63.89_{0.13}$ | $51.81_{0.02}$ | $54.06_{0.04}$ | $47.46_{0.01}$ | $35.56_{0.04}$ | $65.87_{0.01}$ | $84.19_{0.01}$ |
| | RAP | $62.13_{0.09}$ | $74.95_{0.12}$ | $60.94_{0.00}$ | $93.75_{0.00}$ | $63.88_{0.13}$ | $87.75_{0.05}$ | $57.04_{0.05}$ | $81.11_{0.13}$ | $66.64_{0.10}$ | $100_{0.00}$ |
| | Avg | $66.34_{0.06}$ | $60.44_{0.13}$ | $60.57_{0.01}$ | $57.20_{0.02}$ | $66.58_{0.04}$ | $47.62_{0.04}$ | $58.70_{0.03}$ | $57.96_{0.08}$ | $68.45_{0.02}$ | $64.26_{0.14}$ |

Table 6: Evaluation of LLMs in terms of label-wise accuracy on LogicBench(Eval), where $A(Yes)$ and $A(No)$ denote the accuracy for the $Yes$ and $No$ labels, respectively. DRI: Default Reasoning with Irrelevant Information, DRS: Default Reasoning with Several Defaults, DRD: Default Reasoning with a Disabled Default, DRO: Default Reasoning in an Open Domain, RE1: Reasoning about Unknown Expectations I, RE2: Reasoning about Unknown Expectations II, RE3: Reasoning about Unknown Expectations III, RAP: Reasoning about Priorities

an average accuracy of 60.44%, 57.20%, 47.62%, 57.96%, and 64.26%, respectively. Overall, these models display an average performance of $\sim 40\%$, $\sim 61\%$, and $\sim 58\%$ on PL, FOL, and NL.

From Table 6, we can observe that models struggle more with inference rules of PL compared to FOL and NM reasoning. Furthermore, it is noticeable that each model performs relatively better on questions with a negative response (i.e., $No$) compared to questions with a positive response (i.e., $Yes$). This observation suggests that the models struggle to fully comprehend the logical relationship between the context and the conclusion (i.e., lower $A(Yes)$). However, they demonstrate a relatively stronger understanding when the relationship is contradictory in nature (i.e., higher $A(No)$). Moreover, the subsequent section offers a comprehensive analysis of the models' performance on inference rules, as it is crucial to understand their limitations.

## 4.3 Analysis and Discussion

**How do LLMs reason step-by-step?** We investigate the fraction of low-performing axioms that contain various types of logical reasoning steps to predict the answer, and whether the correctness of those steps is correlated with the performance. Here, we perform a case study on ChatGPT. We prompt ChatGPT to generate reasoning steps along with predictions. For PL, we observe that while the model can effectively reason the initial section of the *disjunctive syllogism* involving two possibilities $p$ or $q$, it encounters challenges in deducing whether $q$ should follow from the $\neg p$. For FOL, ChatGPT encounters challenges in comprehending longer logical contexts, resulting in a lack of confidence in establishing the relationship between given propositions. Furthermore, to derive an accurate conclusion when the rules are followed correctly, the model relies on supplementary evidence. We observe that ChatGPT encounters difficulties in comprehending the nuanced meanings of words such as "usually", "normally" and "typically" when establishing sentence relationships within NM reasoning. Notably, when it comes to the rule of default reasoning, ChatGPT fails to grasp inherent associations between two entities that commonly share characteristics. Examples and more analysis of generated explanations for each logic type are presented in App. I.

**Negations are hard to understand when embedded with logical rules.** Regarding PL and FOL, it is apparent that the models struggle more with the DS, DD, and MT inference rules. A closer look at Table 2 reveals that all of these axioms include examples where the models need to draw conclusions based on negated premises. This indicates that the models encounter difficulties when negated premises are introduced. Additionally, the performance of the models tends to decrease when inference rules involve negations.

**Longer inference rules are still challenging.** Table 6 indicates that the models face challenges when handling longer rules, such as BD, CD, and DD, both in PL and FOL. Hence, it can be concluded that these models struggle with longer logical dependencies in the premise, particularly when a higher number of propositions are present. In the case of NM reasoning, the models exhibit lower performance in DRS of NM reasoning, indicating that a higher number of rules in the context often leads to more frequent mistakes.

**Large models are better logical reasoners.** Based on the observed performance from Table 6, it becomes evident that larger model sizes and extensive pre-training data contribute to a better understanding of logical aspects. Consequently, models with larger sizes tend to exhibit higher performance across different types of logic. Nonetheless, the average performance remains at around $52.83\%$, indicating room for improvement in these models' logical comprehension capabilities.

**Effect on other logic datasets** Table 7 represents the accuracy comparison between LogicT5 and baseline T5-large in both single-task and multi-task settings.

| Methods | Models | LogiQA | FOLIO | LogicNLI | ReClor |
|---|---|---|---|---|---|
| Single-Task | T5-large | 16.8 | 69.6 | 82.3 | 35.4 |
| | LogicT5 | **16.9** | **71.2** | **84.4** | **36.8** |
| Multi-Task | T5-large | **21.8** | 83.8 | 68.2 | **42.8** |
| | LogicT5 | 19.7 | **85.6** | **69.8** | 40.0 |

Table 7: Performance comparison between LogicT5 and baseline T5-large in terms of accuracy.

The results indicate that training LLMs on *LogicBench(Aug)* has a greater impact on logic datasets that primarily focus on logical reasoning, such as FOLIO and LogicNLI. Hence, we can observe that LogicT5 consistently outperforms the baseline for LogicT5 and FOLIO. However, LogiQA and ReClor encompass other forms of reasoning in addition to logical reasoning, hence, LogicT5 demonstrates competitive performance on them. More analysis is presented in Appendix M.

## 5 CONCLUSIONS

To study the logical reasoning ability of LLMs, we introduced a novel benchmark called *LogicBench* which consists of 25 distinct inference rules and reasoning patterns covering propositional, first-order, and non-monotonic logics. We released two versions of the dataset: *LogicBench(Eval)* and *LogicBench(Aug)*. *LogicBench(Eval)* serves as a high-quality, cost-effective, and reliable dataset for evaluating LLMs, while *LogicBench(Aug)* can be utilized for training purposes. Through comprehensive experiments, we showed that models such as GPT-3 and ChatGPT do not perform well on *LogicBench*, even though they require the application of only a single inference rule in positive (i.e., label 'Yes') data instance. Furthermore, we demonstrated that LLMs trained using *LogicBench(Aug)* showcase an improved understanding of logical reasoning, resulting in a better performance on existing logic datasets. Though *LogicBench* facilitates the evaluation and improvement of the logical reasoning ability of LLMs, the linguistic diversity of context presented in *LogicBench* can be improved by leveraging LLMs to generate a more story-like context. To this extent, we conducted a preliminary study over PL, and our findings are presented in App. J. This indicates the room for making *LogicBench* even more challenging. Furthermore, *LogicBench* can be further extended by incorporating other inference rules and logic types; and having data instances that require applications of multiple inference rules.

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

## A    EXAMPLE PROMPT FOR SENTENCE GENERATION

Figure 4 illustrates an example prompt for the inference rule, namely, 'modus tollens' from propositional logic (PL). Modus tollens is formally represented as $((p \rightarrow q) \wedge \neg q) \vdash \neg p$, which can be understood in natural language as "If $p$ implies $q$, and we know $\neg q$, then we can conclude $\neg p$." In this prompt, the definition provides a comprehensive description of the inference rule in natural language. To encourage the generation of more relevant and coherent sentences, the prompt includes an examples section that demonstrates how the generated sentences will be utilized in a later stage. This serves, as an illustration, to guide GPT-3 in producing suitable outputs. In Figure 4, we present three examples involving sentences $p$ and $q$, along with their respective contexts and questions. The prompt also includes instructions on how the generated sentences should be formatted.

## B    EXTENDED RELATED WORK

As LLMs such as GPT-4, and Bard continue to evolve rapidly, it becomes increasingly crucial to evaluate their diverse language capabilities, as well as those of forthcoming LLMs. Recently, many datasets have been created that evaluate different language understanding skills such as pronoun resolution (Sakaguchi et al., 2021; Levesque et al., 2012), commonsense reasoning (Talmor et al., 2019), numerical reasoning (Dua et al., 2019; Patel et al., 2021; Mishra et al., 2022), qualitative reasoning (Tafjord et al., 2019b;a), temporal reasoning (Zhou et al., 2019), and feasibility reasoning (Gupta et al., 2022). Now, we present the advancements in prompt and instruction tuning using LLMs.

Figure 4: Example prompt for *Modus Tollens* inference rule from PL.

**Prompt Learning** The introduction of LLMs has significantly shifted the research trend in NLP to prompt-based learning methodologies (Liu et al., 2021b). Many studies have been conducted to investigate the efficacy of prompt-based learning in various applications including Text classification (Yin et al., 2019), Natural Language Inference (NLI) (Schick & Schütze, 2020), and Question Answering (QA) (Jiang et al., 2020), Information Extraction (IE) (Chen et al., 2021; Cui et al., 2021), to name a few. In a recent development, the T0 model employs prompts to achieve zero-shot generalization across various NLP tasks (Sanh et al., 2021). Scao et al. 2021 suggested that the use of prompts could be as valuable as hundreds of data points on average (Le Scao & Rush, 2021).

**Instruction Learning** Efrat et al., 2020 (Efrat & Levy, 2020) was focused on whether existing LLMs understand instructions. The same work in the field of instruction by (Hase & Bansal, 2021; Ye & Ren, 2021; Gupta et al., 2021; Zhong et al., 2021) has been proposed to show that models follow natural language instructions. In addition, Weller et al., 2020 (Weller et al., 2020) developed a framework focusing on NLP systems that solve challenging new tasks based on their description. Mishra et al., 2021 (Mishra et al., 2021b) have proposed natural language instructions for cross-task generalization of LLMs. Similarly, PromptSource (Sanh et al., 2021) and FLAN (Wei et al.,

2021) were built for leveraging instructions and achieving zero-shot generalization on unseen tasks. Moreover, Parmar et al., 2022 (Parmar et al., 2022) shows the effectiveness of instructions in multi-task settings for the biomedical domain. Furthermore, Mishra et al., 2021 (Mishra et al., 2021a) discuss the impact of task instruction reframing. Min et al., 2021 (Min et al., 2021) introduce a framework to better understand in-context learning. Ouyang et al., 2022 (Ouyang et al., 2022) propose the InstructGPT model that is fine-tuned with human feedback to follow instructions. Wang et al., 2022 (Wang et al., 2022a) has developed an instruction-based multi-task framework for few-shot Named Entity Recognition (NER) tasks. In addition, many approaches have been proposed to improve model performance using instructions (Wu et al., 2022; Lin et al., 2021; Wang et al., 2022b; Luo et al., 2022; Kuznia et al., 2022; Patel et al., 2022; Mishra & Nouri, 2022).

**Logic and NLI Datasets** FraCas (Bernardy & Chatzikyriakidis, 2020) offers a unique approach to temporal semantics by converting syntax trees into logical formulas tailored for inference, emphasizing temporal elements such as references, adverbs, aspectual classes, and progressives. The Monotonicity Entailment Dataset (MED) (Yanaka et al., 2019) dives deep into monotonicity reasoning within NLI, probing the synergy between lexical and syntactic structures and spotlighting inherent challenges in both upward and downward monotonic reasoning trajectories. The SICK (Marelli et al., 2014) dataset, with its foundation in 10,000 English sentence pairs, is designed to rigorously evaluate semantic relatedness and entailment, leveraging crowdsourced annotations for precision. HANS, or Heuristic Analysis for NLI Systems (McCoy et al., 2019), stands out by rigorously scrutinizing the dependability of NLI models, putting the spotlight on potential pitfalls tied to syntactic heuristics such as lexical overlap. Lastly, CAD (Vidgen et al., 2021) introduces a meticulously crafted dataset from Reddit entries, targeting the detection of online abuse. This dataset boasts six distinct primary categories, context-aware annotations, provided rationales, and a rigorous group-adjudication methodology ensuring high-quality annotations.

## C    EXAMPLES OF DATA INSTANCES

This section provides examples of *(context, question, answer)* tuples corresponding to each inference rule and reasoning pattern. Additionally, it highlights the diverse range of question variations within the dataset associated with each inference rule and reasoning pattern.

### C.1    WORD CLOUD

Figure 5 provides a word cloud derived from the $LogicBench(Eval)$. This word cloud highlights the logical nature and diversity of our evaluation dataset. Words such as 'if', 'normally', 'usually', and 'then' are prominently featured, suggesting their frequent use in the dataset, and suggesting the logical nature of the dataset. Moreover, we can also observe several words consisting of different ontologies such as 'cat', 'car', 'garden', and many more, suggesting diversity in the dataset.

### C.2    PROPOSITIONAL LOGIC (PL)

Here, we discuss examples of each inference rule present in the PL of the *LogicBench* as shown in Table 8. Table 8 has context related to the inference rule and different variations of the question according to the rule. For instance, the first row of Table 8 shows the example for inference rule, *Hypothetical Syllogism (HS)*, formally expressed as $((p \rightarrow q)) \wedge (q \rightarrow r)) \vdash (p \rightarrow r)$. The context represents the premise, i.e., $((p \rightarrow q)) \wedge (q \rightarrow r))$, and the first question (Q1) represents the conclusion, i.e., $p \rightarrow r$. Hence, Q1 is labeled as "Yes" since it supports the conclusion given the logical context. Furthermore, Q2 to Q4 represent different variations of the question by utilizing the variables $(p, \neg p, r, \neg r)$. For the HS, given the provided context, Q2 to Q4 contain the variations $\neg p \rightarrow r$, $p \rightarrow \neg r$, and $\neg p \rightarrow \neg r$, respectively, and are labeled as "No" since they do not support the conclusion.

### C.3    FIRST-ORDER LOGIC (FOL)

Here, we discuss examples of each inference rule and two axioms (i.e., Existential Instantiation and Universal Instantiation) present in the FOL from the *LogicBench* as shown in Table 9. *Existential Generalization (EG)*, formally expressed as $P(a) \Rightarrow \exists x P(x)$ indicates that there is an element $a$

r

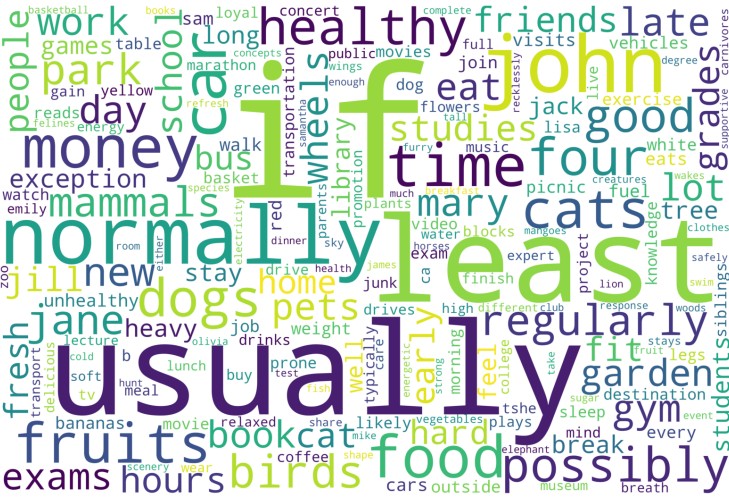

Figure 5: Word cloud of context present in the LB(eval)

in the domain for which $P(a)$ is true, then we know that $\exists x P(x)$ is true. *Universal Instantiation* formally expressed as $\forall x\, A \Rightarrow A\{x \mapsto a\}$ indicates that a statement holds true for all instances (x) within a specific category $A$, hence it is also true for specific instance $a$.

Table 9 represents context related to the inference rule and variations of the question. The process of generating data instances for FOL follows a similar approach to that of PL. For example, the first row of Table 9 shows the example for axiom, *Existential Instantiation (EI)*, formally expressed as $\exists x P(x) \Rightarrow P(a)$. The context represents the initial premise $\exists x P(x)$ and the first question (Q1) represents the conclusion, i.e., $P(a)$. Hence, Q1 is labeled as "Yes" since it supports the conclusion given the logical context. Furthermore, we generate the only variant of the question based on $\neg P(a)$ and labeled it as $No$ since it does not support the conclusion.

### C.4 NON-MONOTONIC (NM) REASONING

Here, we discuss examples of each reasoning pattern present in the NM reasoning from the *LogicBench* as shown in Table 10. Table 10 has context related to the reasoning pattern and different variants of the question. For example, the first row of Table 10 shows the example for *Default Reasoning with Irrelevant Information (DRI)*. For this reasoning, based on the given context, there are also two possible variations of the question where one with a correct conclusion labeled as $Yes$ and another with an incorrect conclusion labeled as $No$.

## D EXAMPLES OF NL CONVERSION

This section illustrates the way natural language logical context and questions are created using the generated sentence in Stage 1. Table 11 shows examples of how context and question are generated from sentences corresponding to each inference rule for PL and FOL. Similarly, Table 12 shows examples of NM reasoning. From Table 11, we can see an example of sentence pairs $(p, q)$ and their corresponding negation pairs $(\neg p, \neg q)$ for the 'modus tollens' inference rule from PL. These pairs are utilized to generate logical context and questions. Similarly, in the second row, we have four generic rules with variable $x$ $(p(x), q(x), r(x), s(x))$ and their specific cases (i.e., $x = a$), along with their respective negative sentence pairs $[(p(a), \neg p(a)), (q(a), \neg q(a)), (r(a), \neg r(a)), (s(a), \neg s(a))]$. These examples demonstrate the generation of logical context and questions for the FOL inference rule called 'Bidirectional Dilemma (BD)', as shown in Table 11. From Table 12, the first row presents an example of context and questions generated from a sentence pair for the 'Default Reasoning with Irrelevant Information (DRI)' from NM reasoning. In this specific instance, the generated sentences are $(p, q, r, s, t)$, and the negation is only required for the sentence $t$. Therefore, there is a single negation pair $(t, \neg t)$, which is used to generate questions specific to the 'DRI'.

| Rule | Context | Question |
|------|---------|----------|
| HS | If Jim cleaned his room, then he will get a reward. If he will get a reward, then he will buy a new toy. | **Q1:** If Jim cleaned his room, does this imply that he will buy a new toy? (Yes)
**Q2:** If Jim didn't clean his room, does this entail that he won't buy a new toy? (No)
**Q3:** If Jim cleaned his room, does this imply that he won't buy a new toy? (No)
**Q4:** If Jim didn't clean his room, does this imply that he will buy a new toy? (No) |
| DS | We know that at least one of the following is true (1) Chloe is studying for her exams and (2) Mila is going on vacation. Note that we do not know which ones of (1) and (2) are true. It might be the case that only (1) is true, or only (2) is true or both are true. | **Q1:** If Chloe isn't studying for her exams, does this entail that Mila is going on vacation? (Yes)
**Q2:** If Chloe isn't studying for her exams, does this mean that Mila isn't going on vacation? (No)
**Q3:** If Chloe is studying for her exams, does this imply that Mila isn't going on vacation? (No)
**Q4:** If Chloe is studying for her exams, does this imply that Mila is going on vacation? (No) |
| CD | If I go for a walk, then I will get some fresh air. If I stay home, then I will watch a movie. We know that at least one of the following is true (1) I go for a walk and (2) I stay home. Note that we do not know which ones of (1) and (2) are true. It might be the case that only (1) is true, or only (2) is true or both are true. | If at least one of (1) and (2) is true, can we say, at least one of the following must always be true?
**Q1:** (a) I will get some fresh air or (b) I will watch a movie (Yes)
**Q2:** (a) I won't get some fresh air and (b) I will watch a movie (No)
**Q3:** (a) I will get some fresh air and (b) I won't watch a movie (No)
**Q4:** (a) I won't get some fresh air and (b) I won't watch a movie (No) |
| DD | If I order takeout, then I will save time. If I cook a meal, then I will save money. We know that at least one of the following is true (1) I won't save time and (2) I won't save money. Note that we do not know which ones of (1) and (2) are true. It might be the case that only (1) is true, or only (2) is true or both are true. | If at least one of (1) and (2) is true, can we say, at least one of the following must always be true?
**Q1:** (a) I don't order takeout or (b) I don't cook a meal (Yes)
**Q2:** (a) I order takeout and (b) I cook a meal (No)
**Q3:** (a) I don't order takeout and (b) I cook a meal (No)
**Q4:** (a) I order takeout and (b) I don't cook a meal (No) |
| BD | If it rains, then we will stay inside. If it is sunny, then we will go for a walk. We know that at least one of the following is true (1) it rains and (2) we will not go for a walk. Note that we do not know which ones of (1) and (2) are true. It might be the case that only (1) is true, or only (2) is true or both are true. | If at least one of (1) and (2) is true, can we say, at least one of the following must always be true?
**Q1:** (a) we will stay inside or (b) it is not sunny (Yes)
**Q2:** (a) we will not stay inside and (b) it is sunny (No)
**Q3:** (a) we will stay inside and (b) it is sunny (No)
**Q4:** (a) we will not stay inside and (b)it is not sunny (No) |
| MT | If Mason left his job, then he will not receive any salary. | **Q1:** If he will receive any salary, does this mean that Mason didn't leave his job? (Yes)
**Q2:** If he will receive any salary, does this mean that Mason left his job? (No)
**Q3:** If he will not receive any salary, does this imply that Mason didn't leave his job? (No)
**Q4:** If he will not receive any salary, does this mean that Mason left his job? (No) |
| MI | If Rohan forgot his lunch, then he will not eat at school. | Based on context, can we say, at least one of the following must always be true?
**Q1:** (a) Rohan didn't forget his lunch and (b) he will not eat at school (Yes)
**Q2:** (a) Rohan forgot his lunch and (b) he will eat at school (No)
**Q3:** (a) Rohan forgot his lunch and (b) he will not eat at school (No)
**Q4:** (a) Rohan didn't forget his lunch and (b) he will eat at school (No) |
| CT | We know that at least one of the following is true (1) Tom is an avid reader and (2) he devours books of all genres. Note that we do not know which ones of (1) and (2) are true. It might be the case that only (1) is true, or only (2) is true or both are true. | If at least one of (1) and (2) is true, can we say, at least one of the following must always be true?
**Q1:** (a) he devours books of all genres or (b) Tom is an avid reader (Yes)
**Q2:** (a) he doesn't devour books of all genres and (b) tom is an avid reader (No)
**Q3:** (a) he devours books of all genres and (b) tom isn't an avid reader (No)
**Q4:** (a) he doesn't devour books of all genres and (b) tom isn't an avid reader (No) |

Table 8: Examples of context and question-answer pairs for each axiom of *Proportional logic* from the LogicBench; HS: Hypothetical Syllogism, DS: Disjunctive Syllogism, CD: Constructive Dilemma, DD: Destructive Dilemma, BD: Bidirectional Dilemma, MT: Modus Tollens, MI: Material Implication, CT: Commutation

| Rule | Context | Question |
|------|---------|----------|
| UI | All students need to take an exam to complete their degree. Reema is a student. | **Q1:** Does Reema need to take an exam to complete her degree? (Yes)
**Q2:** Does Reema need not to take an exam to complete her degree? (No) |
| EG | James won the marathon race | **Q1:** Does this imply that someone won the marathon race? (Yes)
**Q2:** Does this mean that no one won the marathon race? (No) |
| MP | If someone is exhausted, then they will take a rest. | **Q1:** If Jack is exhausted, does this entail that he will take a rest? (Yes)
**Q2:** If Jack isn't exhausted, does this imply that he won't take a rest? (No)
**Q3:** If Jack is exhausted, does this entail that he won't take a rest? (No)
**Q4:** If Jack isn't exhausted, does this entail that he will take a rest? (No) |
| HS | If someone buys all the necessary supplies, then they can start the project. If they can start the project, then they will finish it on time. | **Q1:** If Lily bought all the necessary supplies, does this mean that she will finish it on time? (Yes)
**Q2:** If Lily didn't buy all the necessary supplies, does this imply that she won't finish it on time? (No)
**Q3:** If Lily bought all the necessary supplies, does this entail that she won't finish it on time? (No)
**Q4:** If Lily didn't buy all the necessary supplies, does this imply that she will finish it on time? (No) |
| DS | We know that at least one of the following is true (1) they can go to a museum and (2) they can visit a park. Note that we do not know which ones of (1) and (2) are true. It might be the case that only (1) is true, or only (2) is true or both are true. | **Q1:** If Jill can't go to a museum, does this imply that she can visit a park? (Yes)
**Q2:** If Jill can't go to a museum, does this entail that she can't visit a park? (No)
**Q3:** If Jill can go to a museum, does this entail that she can't visit a park? (No)
**Q4:** If Jill can go to a museum, does this imply that she can visit a park? (No) |
| CD | If someone is painting a picture, then they will frame it. If they are writing a story, then they will publish it. We know that at least one of the following is true (1) John is painting a picture. and (2) She is writing a story. Note that we do not know which ones of (1) and (2) are true. It might be the case that only (1) is true, or only (2) is true or both are true. | If at least one of (1) and (2) is true, can we say, at least one of the following must always be true?
**Q1:** (a) He will frame it. and (b) She will publish it. (Yes)
**Q2:** (a) he won't frame it. and (b) She will publish it. (No)
**Q3:** (a) He will frame it. and (b) she won't publish it. (No)
**Q4:** (a) he won't frame it. and (b) she won't publish it. (No) |
| DD | If someone takes care of her health, then they will be fit and healthy. If they indulge in unhealthy habits, then they will be prone to diseases. We know that at least one of the following is true (1) Jenny won't be fit and healthy and (2) she won't be prone to diseases. Note that we do not know which ones of (1) and (2) are true. It might be the case that only (1) is true, or only (2) is true or both are true. | If at least one of (1) and (2) is true, can we say, at least one of the following must always be true?
**Q1:** (a) Jenny doesn't take care of her health and (b) she doesn't indulge in unhealthy habits (Yes)
**Q2:** (a) Jenny takes care of her health and (b) she indulges in unhealthy habits (No)
**Q3:** (a) Jenny doesn't take care of her health and (b) she indulges in unhealthy habits (No)
**Q4:** (a) Jenny takes care of her health and (b) she doesn't indulge in unhealthy habits (No) |
| BD | If someone drinks lots of water, then they will feel hydrated. If they eat too much sugar, then they will experience a sugar crash. We know that at least one of the following is true (1) Jane drinks lots of water and (2) she won't experience a sugar crash. Note that we do not know which ones of (1) and (2) are true. It might be the case that only (1) is true, or only (2) is true or both are true. | If at least one of (1) and (2) is true, can we say, at least one of the following must always be true?
**Q1:** (a) she will feel hydrated and (b) she doesn't eat too much sugar (Yes)
**Q2:** (a) she won't feel hydrated and (b) she eats too much sugar (No)
**Q3:** (a) she will feel hydrated and (b) she eats too much sugar (No)
**Q4:** (a) she won't feel hydrated and (b) she doesn't eat too much sugar (No) |
| MT | If someone visits the park, then they have to wear a mask. | **Q1:** If he doesn't have to wear a mask, does this imply that John doesn't visit the park? (Yes)
**Q2:** If he doesn't have to wear a mask, does this entail that John visits the park? (No)
**Q3:** If he has to wear a mask, does this imply that John doesn't visit the park? (No)
**Q4:** If he has to wear a mask, does this imply that John visits the park? (No) |

Table 9: Examples of context and question-answer pairs for each axiom of *First order logic* from the LogicBench.

| Rule | Context | Question |
|------|---------|----------|
| DRI | Cats and dogs are mammals. mammals typically have fur. cats don't have fur. dogs are loyal animals. | **Q1:** Does this imply that dogs have fur? (Yes)
**Q2:** Does this entail that dogs don't have fur? (No) |
| DRS | John and Mary are parents. parents are usually loving and supportive. parents are normally responsible. Mary isn't loving and supportive. John is not responsible. | **Q1:** Does this imply that Mary is responsible and John is loving and supportive? (Yes)
**Q2:** Does this entail that Mary isn't responsible and John is loving and supportive? (No)
**Q3:** Does this imply that Mary is responsible and John isn't loving and supportive? (No)
**Q4:** Does this entail that Mary isn't responsible and John isn't loving and supportive? (No) |
| DRD | Jenny and Anna are tall. tall people usually play basketball. Anna is possibly an exception to this rule. | **Q1:** Does this entail that Jenny plays basketball? (Yes)
**Q2:** Does this mean that Jenny doesn't play basketball? (No) |
| DRO | Hummingbirds are birds. Birds migrate south for the winter. Hummingbirds do not migrate south for the winter. | **Q1:** Does this mean that all other birds than hummingbirds migrate south for the winter? (Yes)
**Q2:** Does this mean that all other birds than hummingbirds don't migrate south for the winter? (No) |
| RE1 | Cats, dogs, and horses are animals. animals are usually considered to be intelligent creatures. at least one of the cats or dogs is not considered intelligent. | **Q1:** Does this entail that horses are considered to be intelligent creatures and exactly one of the cats or dogs is not considered intelligent? (Yes)
**Q2:** Does this mean that horses aren't considered to be intelligent creatures and exactly one of cats or dogs is not considered intelligent? (No)
**Q3:** Does this mean that horses are considered to be intelligent creatures and exactly one of cats or dogs is considered intelligent? (No)
**Q4:** Does this implies that horses aren't considered to be intelligent creatures and exactly one of cats or dogs is considered intelligent? (No) |
| RE2 | Cats normally meow. at least one species of cat doesn't meow. | **Q1:** Does this entail that exactly one species of cat doesn't meow? (Yes)
**Q2:** Does this imply that exactly one species of cat meows? (No) |
| RE3 | Cars have four wheels. wheels normally have spokes. at least one wheel does not have spokes. | **Q1:** Does this imply that cars have four wheels with spokes? (Yes)
**Q2:** Does this mean that cars don't have four wheels with spokes? (No) |
| RAP | John asserts that Sally was in the store. Jane asserts that Sally was not in the store. | **Q1:** If John's evidence is more reliable than Jane's, does this mean that Sally was in the store? (Yes)
**Q2:** If John's evidence is more reliable than Jane's, does this mean that Sally wasn't in the store? (No)
**Q3:** If John's evidence is less reliable than Jane's, does this entail that Sally was in the store? (No)
**Q4:** If John's evidence is less reliable than Jane's, does this imply that Sally wasn't in the store? (Yes) |

Table 10: Examples of context and question-answer pairs for each axiom of *Non-monotonic logic* from the LogicBench.

# E  FEW-SHOT EXPERIMENTS

This section discusses the performance of the different LLMs in a few-shot setting on the *Log-icBench(Eval)*. Here, we provide a prompt along with four distinct examples (two examples with $Yes$ and two examples with $No$). For the fair comparison with Table 6, we analyze an average performance across $A(Yes)$. Table 13 shows the performance for each inference rule and reasoning patterns achieved by FLAN-T5, Tk-instruct, GPT-3, ChatGPT, and GPT-4.

From Table 13, we can observe that in-context examples are helpful for GPT-3 and GPT-4 since both models consistently outperform zero-shot baselines by large margins in terms of $A(Yes)$. In particular, GPT-3, and GPT-4 improve performance by $\sim 26\%$ and $\sim 30\%$ (average across all logic) compared to zero-shot baseline, respectively. GPT-4 has been shown to be remarkably good at following the in-context exemplars and mimicking the process to reach the correct conclusions. Thus, leveraging the in-context exemplars, GPT-4 achieves high accuracy in a few-shot setting. As suggested in (Lu et al., 2022), prompt and instruction-tuned models are sensitive to in-context examples. Hence, we see performance variations in Table 13 across all models. Specifically, FLAN-T5 improves an average performance by $\sim 4\%$ for FOL, however, it shows competitive performance on PL and NM reasoning. Furthermore, Tk-instruct improves an average performance by $\sim 9\%$ for PL and $\sim 4\%$ for NM, however, it shows a performance drop on FOL by $\sim 6\%$. Interestingly, in-context examples in a few-shot setting hamper the performance of ChatGPT by $\sim 6\%$ for PL, compared to zero-shot. However, ChatGPT improves performance by $\sim 4\%$, and $\sim 23\%$ on FOL and NM reasoning, respectively. Improved performance in NM reasoning demonstrates that the inclusion of in-context examples enhances the ability of these models to comprehend the nuanced meanings of logical terms such as "usually" and "typically".

# F  EVALUATION OF LOGICBENCH WITHOUT CHAIN-OF-THOUGHT

In this section, we discuss experiments carried out in a zero-shot setting without a chain of thought (CoT). All the experiments conducted in the zero-shot setting were performed using three distinct prompts. The reported results in Table 14 represent the average performance across these prompts. The following are the three different prompts utilized in the experiments:

Prompt 1:

> Given the context and question, respond in 'yes' or 'no'.

Prompt 2:

> Answer the given question ONLY in 'yes' or 'no' using logical reasoning ability. DO NOT generate anything as an answer apart from 'yes' and 'no'.

Prompt 3:

> Given context contains rules of logical reasoning in natural language. Answer the given question based on context ONLY in 'yes' or 'no' using logical reasoning ability. DO NOT generate anything as an answer apart from 'yes' and 'no'.

From Table 14, we can observe that zero-shot CoT prompting (Table 6) consistently improves the performance of FLAN-T5 and Tk-instruct models. Interestingly, GPT-4, GPT-3, and ChatGPT exhibit competitive performance or performance drop when employing CoT-based prompting. It is well-established that CoT generally enhances reasoning task performance (Wei et al., 2022); however, we observed that these models encounter difficulties when dealing with intricate logical rules. Consequently, enabling them to generate rationales can sometimes result in erroneous predictions. However, in the absence of CoT, these models may resort to employing simple heuristics to answer questions rather than following a logical reasoning chain.

## G ANALYSIS ON BARD AND LLaMa-2

**Bard**    This section discusses a case study carried out on a Bard (Google) with the subset of the *LogicBench* dataset for each inference rule and reasoning pattern from PL, FOL, and NM reasoning. To evaluate this model, we use below prompt:

> Given context contains rules of logical reasoning in natural language. Answer the given question based on context ONLY in 'yes' or 'no' using logical reasoning ability. DO NOT generate anything as an answer apart from 'yes' and 'no'.

Due to the unavailability of Bard developer API during the evaluation timeline of this paper (August 2023), we manually evaluate a carefully selected subset of *LogicBench(Eval)*. We randomly selected five data samples containing *(context, question, 'Yes')* triplets since the goal of this evaluation is to see if the model can identify the relationship between logical rules (context) and conclusion (question). The experiment was conducted on a total of 125 samples by combining samples from all 25 inference rules and axioms. Results are presented in Table 15. Bard performs well on the FOL with the highest $A(Yes)$ of 71.1% while it achieves $A(Yes)$ of 65%, and 15% on PL and NM reasoning, respectively. Bard performs poorly on NM reasoning showing that it struggles on understanding the nuance of logical words such as 'normally', 'usually', and 'typically'.

| Logic | Bard |
|---|---|
| PL | 65.0% |
| FOL | 71.1% |
| NM | 15.0% |
| Average | 51.2% |

Table 15: Case study of performance of BARD on a subset of *LogicBench*

**LLaMa-2**    We evaluate LLaMa-2-7B model on LogicBench(Eval) using Zero-shot-CoT. All the experiments conducted in the zero-shot setting were performed using three distinct prompts presented in Appendix H.2. The reported results in Table 16 represent the average performance across these prompts. From the results, we can observe that LLaMa-2 also exhibits similar behavior as other LLMs (Table 6) in terms of performance on LogicBench(Eval).

## H EXPERIMENTAL SETUP

### H.1 EXTENDED DISCUSSION ON EXPERIMENTS

**Zero-shot setting**    We evaluate GPT-4, GPT-3(text-davinci-003) and ChatGPT by utilizing their APIs provided by OpenAI[3]. The evaluation is conducted on the versions of GPT-4, GPT-3, and ChatGPT released in August 2023. It's important to note that these models are regularly updated, so when reproducing the results presented in Table 6 (main paper), there is a possibility of variations. For FLAN-T5, and Tk-instruct, we utilize the 3B, and 3B versions, respectively, from the huggingface model repository[4].

**Experiments on other logic datasets**    In single and multi-task experiments on other logic datasets, we fine-tune the T5-large model for 10 epochs with a batch size of 16, 1024 maximum input length, an adaptive learning rate of $5e-05$, and an AdamW optimizer for each experiment. All experiments are performed using NVIDIA RTX A6000 GPUs.

### H.2 PROMPTS

All the experiments conducted in the zero-shot setting were performed using three distinct prompts. The reported results in Table 6 (main paper) represent the average performance across these prompts. All the prompts follow the common pattern which includes task description and formatting instructions. The following are the three different prompts utilized in the experiments:

---

[3]https://platform.openai.com/docs/guides/gpt
[4]https://huggingface.co/models

**Prompt 1:**

> Given the context that contains rules of logical reasoning in natural language and question, perform step-by-step reasoning to answer the question. Based on context and reasoning steps answer the question ONLY in 'yes' or 'no'. Please use the below format:
> **Context:** [text with logical rules]
> **Question:** [question that is based on context]
> **Reasoning steps:** [generate step-by-step reasoning]
> **Answer:** Yes/No

**Prompt 2:**

> Given the context and question, think step-by-step logically to answer the question. Answer the question ONLY in 'yes' or 'no'. Please use the below format:
> **Context**: [text with logical rules]
> **Question**: [question that is based on context]
> **Reasoning steps**: [generate step-by-step reasoning]
> **Answer**: Yes/No

**Prompt 3:**

> Let's think step-by-step to answer the question given context. Answer the question ONLY in 'yes' or 'no'. Please use the below format:
> **Context**: [text with logical rules]
> **Question**: [question that is based on context]
> **Reasoning steps**: [generate step-by-step reasoning]
> **Answer**: Yes/No

## I  CASE STUDY ON LOGICAL EXPLANATIONS

This section represents a case study carried out on an inference rule of each type of logic where ChatGPT is not able to perform well. We use examples generated by *Prompt 1* to carry out this manual analysis. Table 17, 18, and 19 represents a case study for *Disjunctive Syllogism (DS)* from PL, *Destructive Dilemma (DD)* from FOL, and *Default Reasoning with Several Default (DRS)* from NM, respectively. From Table 6 (main paper), we can observe that ChatGPT shows poor performance on these inference rules and reasoning patterns, hence, we believe that analysis of logical explanations corresponding to these can give us more insights into the performance of ChatGPT. Here, we prompt the ChatGPT model using *Prompt 1* to generate reasoning steps along with a predicted answer. Table 17, 18, and 19 represents five examples of *(Context, Question, Correct answer, Logical reasoning steps)* pairs generated by ChatGPT.

## J  INVESTIGATING LINGUISTIC DIVERSITY

In Stage 2 of data creation (§3.2.2), *LogicBench(Eval)* utilizes pre-defined templates to create natural language (NL) context, ensuring each context adheres to logical rule. To further improve the linguistic diversity of those instances, we leverage GPT-3 to generate a more story-like context, making the language within the context appear more human-like. Here, this case study only focuses on inference rules in PL.

For instance, in the context created by *LogicBench* for the Modus Tollens inference rule is "If Liam finished his work early, then he will order pizza for dinner." Subsequently, we prompt the GPT-3 model to convert this context into a more human-like narrative. Now, the newly generated context is "Liam had been diligently working on a project all day, determined to complete it before dinner. As he neared the project's end, he indeed finished it early. Recognizing his well-earned respite, he decided to treat himself to a pizza dinner as a reward."

Table 20 presents three distinct examples of generated stories that adhere to logical rules given the original context. Below is the prompt used to instruct GPT-3 in generating these stories:

> **Rule**: [provide rule formulation]
> **Context**: [original context from LogicBench]
>
> Improve the context in human-like language and make a small story around it. To generate a story, DO NOT change the main character from the context. Make sure that the story adheres to the logical rule presented above. Generate only one paragraph story.

Here, we evaluate the GPT-3(davinci-003) model given *Prompt 1* from §H.2 in original and story-like context. It is important to note that only the context has been modified, while the questions remain consistent with those in *LogicBench(Eval)*. Table 21 provides the performance of GPT-3 using original and story-like context. The findings indicate a noticeable drop in GPT-3's performance when the context becomes more human-like. This drop suggests that enhancing the human-like nature of the context introduces added complexity to the *LogicBench(Eval)*, providing room to further extend our dataset. However, we believe that this more natural context may also become more susceptible to not adhering to logical correctness, requiring manual verification. In contrast, the template-based creation approach employed in *LogicBench(Eval)* ensures logical correctness in the context.

## K  CROSS-TYPE GENERALIZATION

In this section, we delve into assessing the language model's capability in achieving cross-logic-type generalization, employing T5-Large. Our primary focus is to scrutinize how effectively the model, when fine-tuned extensively on PL, can adapt its knowledge to other logic types (i.e., FOL and NM).

To conduct our experiments, we initially fine-tuned T5-Large separately on PL, FOL, and NM using LogicBench(Aug). Subsequently, we evaluated the model's performance after fine-tuning PL by testing it on FOL and NL, and vice versa using LogicBench(Eval). Our fine-tuning process involved training the T5-Large model for 3 epochs with a batch size of 16, a maximum input length of 1024 tokens, an adaptive learning rate set to $5e-05$, and utilizing the AdamW optimizer. These experiments were executed on NVIDIA RTX A6000 GPUs.

| Model Training | Model Evaulation | | |
|---|---|---|---|
| | **PL** | **FOL** | **NM** |
| PL | 93.13 | 86.25 | 49.09 |
| FOL | 86.72 | 97.34 | 54.73 |
| NM | 43.28 | 57.66 | 92.73 |

Table 22: Results on cross-type logic using T5-Large.

Table 22 shows the results of cross-task generalization. Experimental results reveal that the model fine-tuned on PL performs fairly well on FOL ($\sim 86\%$), though it remains lower than the supervised model fine-tuned on FOL ($\sim 97\%$). A similar observation is made for the model fine-tuned on FOL which fairly does well on PL. However, both these models struggle to generalize on non-classical NM reasoning showing lower accuracy ($\sim 52\%$). Additionally, the model fine-tuned on NM reasoning struggles in generalizing to classical logic, PL, and FOL.

## L  SMALL-SCALE HUMAN EVALUATION ON LOGICBENCH(EVAL)

To further support the integrity and reliability of the benchmark, we hired three graduate student volunteers to manually check the quality of generated data instances. We randomly selected 5 instances from each inference rule, resulting in a total of 125 instances across 25 reasoning patterns for human evaluation. In particular, annotators are asked to provide binary answers (yes/no) for "Validity of generated context", and "Validity of generated question" to make sure they adhere to the intended logical structure. Each instance is annotated by three different annotators. The inter-annotator agreement, measured with raw/observed agreement is 0.956. When there was a disagreement between annotators, a majority class was preferred.

**Questions provided to annotators**

- Validity of context: Is the inference rule followed in the context properly? (Answer Choice: Yes or No)

- Validity of question: Does the question include the entailment component of the inference rule and answer align correctly with it? (Answer Choice: Yes or No)

**Task Instruction (Example of Modus Ponens - FOL):**

Here is the task instruction example provided to annotators for Modus Ponens. Similar task instructions are provided for other inference rules as well.

> **Rule**: Modus Ponens
> **Mathematical rule/generalized formula**: $((p \rightarrow q) \land p) \vdash q$
> **Set of propositions:** p and q
> **Good Example:**
>
> > Context: If someone is exhausted, then they will take a rest.
> > Question: If Jack is exhausted, does this entail that he will take a rest?
> > Answer: yes
> > Explanation:
> > To analyze the above example, we will use the asked question to assess the quality of the sample.
> > 1. The context has the assumptions as $p \rightarrow q$ - someone is exhausted implies they will take rest and hence the context follows the assumptions correctly. Thus, the answer to the first question is 'yes'.
> > 2. No we know that Jack is exhausted (p) and the question states "Does this entail that he will take rest?" Here, he will take rest indicates entailment part of the inference rule. Furthermore, the answer provided based on a question is correct since it follows the Modus Ponens rule. Thus, the answer to the second question is 'yes'.

Along with this instruction, we provide randomly selected 5 instances of Modus Ponens for annotation (similar to other inference rules). Authors closely monitored the annotation process by addressing annotators' queries regarding their understanding of task instructions[5].

## M   FURTHER DISCUSSION ON RESULTS

**Lower performance of GPT-4 on PL as compared to NM**   In the development of AI, non-monotonic logics were partly developed to formalize natural language constructs, such as "normally birds fly", that were not formalizable in a straightforward manner using classical mathematical logics. Thus, while it was difficult for researchers to come up with non-monotonic logics and formalize non-monotonic reasoning, the fact that they were usually motivated by natural language examples, suggests that many of the non-monotonic reasoning aspects are present in the NL text in the wild that is used in the pre-training of the ultra-large LLMs such as GPT4. On the other hand, some of the PL features are counterintuitive to humans such as if we have contradiction (a and a) then everything (even unrelated) is entailed. Also, some PL features are perhaps less prevalent in human writing (on which LLMs are trained) - such as Modes Tollens. Table 6 shows that GPT-4 achieves $\sim 97\%$ accuracy (A(yes)) for simple/straightforward inference rules such as MP(FOL) and HS(PL). However, GPT-4 performance dropped to $\sim 85\%$ A(Yes) for MT. As the complexity of inference rules increases such as BD, CD, and DD (formal expression presented in Table 2), GPT-4 performance further deteriorates (Table 6). Thus, the evaluations support the hypothesis about the frequency of such reasoning patterns in the "training" data.

---

[5]Further details are presented at `https://anonymous.4open.science/r/LogicBench-EEBB/human_eval/readme.md`

**Performance of GPT-4 on PL as compared to FOL**    Results in Table 6 show an average improvement in FOL results because of LLMs' high accuracy on two axioms, EI and UI. However, when we compare the performance across the six common inference rules between PL and FOL (MT, HS, DS, BD, CD, DD), GPT-4 achieves an average of 49.16% A(Yes) for PL and 39.54% A(Yes) for FOL which shows that our results show the expected behavior. The high accuracy of GPT-4 in handling EI and UI can be attributed to their simplicity. While from human experience and complexity theory, FOL is harder than PL in general; in the LLM context, the crucial factor becomes what kind of logical sentences LLMs are pre-trained on. It seems that LLMs are pre-trained more on simple FOL sentences than on simple PL sentences.

| Rule | Generate Sentences in Step 1 | NL logical expressions |
|---|---|---|
| MT | p: Liam finished his work early.
∼p: Liam did not finish his work early.
q: He will order pizza for dinner.
∼q: He will not order pizza for dinner. | **Context:** If Liam finished his work early, then he will order pizza for dinner.

**Question:** If he won't order pizza for dinner, does this imply that Liam didn't finish his work early? |
| BD | p(x): someone drinks lots of water
q(x): they will feel hydrated
r(x): they eat too much sugar
s(x): they will experience a sugar crash
p(a): Jane drinks lots of water
∼p(a): Jane does not drink lots of water
q(a): she will feel hydrated
∼q(a): she will not feel hydrated
r(a): she eats too much sugar
∼r(a): she does not eat too much sugar
s(a): she will experience a sugar crash
∼s(a): she will not experience a sugar crash | **Context:** If someone drinks lots of water, then they will feel hydrated. If they eat too much sugar, then they will experience a sugar crash. We know that at least one of the following is true (1) Jane drinks lots of water and (2) she won't experience a sugar crash. Note that we do not know which ones of (1) and (2) are true. It might be the case that only (1) is true, or only (2) is true or both are true.

**Question:** If at least one of (1) and (2) is true, can we say, at least one of the following must always be true? (a) she will feel hydrated and (b) she doesn't eat too much sugar. |
| MP | p(x): someone is exhausted.
q(x): they will take a rest.
p(a): Jack is exhausted.
∼p(a): Jack is not exhausted.
q(a): he will take a rest.
∼q(a): he will not take a rest. | **Context:** If someone is exhausted, then they will take a rest.

**Question:** If Jack is exhausted, does this entail that he will take a rest? |
| DS | p: John is watching a movie.
∼p: John is not watching a movie.
q: He is playing a game.
∼q: He is not playing a game. | **Context:** We know that at least one of the following is true (1) John is watching a movie and (2) he is playing a game. Note that we do not know which ones of (1) and (2) are true. It might be the case that only (1) is true, or only (2) is true or both are true.

**Question:** If he is not watching a movie, does this mean that is playing a game? |
| HS | p(x): someone buys all the necessary supplies.
q(x): they can start the project.
r(x): they will finish it on time.
p(a): Lily bought all the necessary supplies.
∼p(a): Lily did not buy all the necessary supplies.
q(a): She can start the project.
∼q(a): She can not start the project.
s(a): She will finish it on time.
∼s(a): She will not finish it on time. | **Context:** If someone buys all the necessary supplies, then they can start the project. If they can start the project, then they will finish it on time.

**Question:** If Lily didn't buy all the necessary supplies, does this imply that she won't finish it on time? |
| CD | p: Harry goes to the park.
∼p: Harry does not go to the park.
q: he will have a picnic with his family
∼q: he will not have a picnic with his family
r: he goes to the beach
∼r: he does not go to the beach
s: he will swim in the ocean
∼s: he will not swim in the ocean | **Context:** If Harry goes to the park, then he will have a picnic with his family. If he goes to the beach, then he will swim in the ocean. We know that at least one of the following is true (1) Harry goes to the park and (2) he goes to the beach. Note that we do not know which ones of (1) and (2) are true. It might be the case that only (1) is true, or only (2) is true or both are true.

**Question:** If at least one of (1) and (2) is true, can we say, at least one of the following must always be true? (a) he will have a picnic with his family or (b) he will swim in the ocean |
| DD | p: I order takeout
∼p: I did not order takeout
q: I will save time
∼q: I will not save time
r: I cook a meal
∼r: I did not cook a meal
s: I will save money
∼s: I will not save money | **Context:** If I order takeout, then I will save time. If I cook a meal, then I will save money. We know that at least one of the following is true (1) I won't save time and (2) I won't save money. Note that we do not know which ones of (1) and (2) are true. It might be the case that only (1) is true, or only (2) is true or both are true.

**Question:** If at least one of (1) and (2) is true, can we say, at least one of the following must always be true? (a) I order takeout and (b) I cook a meal |
| CT | p: Tom is an avid reader
∼p: Tom is not an avid reader
q: he devours books of all genres
∼q: he does not devour books of all genres | **Context:** We know that at least one of the following is true (1) Tom is an avid reader and (2) he devours books of all genres. Note that we do not know which ones of (1) and (2) are true. It might be the case that only (1) is true, or only (2) is true or both are true.

**Question:** If at least one of (1) and (2) is true, can we say, at least one of the following must always be true? (a) he devours books of all genres or (b) Tom is an avid reader |
| MI | p: he is not eating healthy
∼p: he is eating healthy
q: he will not gain weight
∼q: he will gain weight | **Context:** If he is not eating healthy, then he will not gain weight.

**Question:** Based on context, can we say, at least one of the following must always be true? (a) he is eating healthy and (b) he will gain weight |
| EG | p(x): someone has coding skills
a: Sheila
p(a): Sheila is a proficient programmer | **Context:** sheila is a proficient programmer

**Question:** does this mean that someone has coding skills? |
| UI | p(x): students need to take an exam to complete their degree
a: Reema
p(a): Reema is a student. | **Context:** All students need to take an exam to complete their degree. Reema is a student.

**Question:** Does Reema need to take an exam to complete her degree |

Table 11: Illustrative examples of NL logical context and questions created using sentences that are generated in §3.2.1 for axioms covered in PL and FOL.

| Rule | Generate Sentences in Step 1 | NL logical expressions |
|---|---|---|
| DRI | p: Cats and dogs are mammals. 
 q: Mammals typically have fur. 
 r: Cats don't have fur. 
 s: Dogs are loyal animals. 
 t: Dogs have fur. 
 ∼t: Dogs don't have fur. | **Context:** Cats and dogs are mammals. mammals typically have fur. cats don't have fur. dogs are loyal animals. 

 **Question:** Does this imply that dogs have fur? |
| DRS | p: John and Mary are parents. 
 q: Parents are usually loving and supportive. 
 r: Parents are normally responsible. 
 s: Mary isn't loving and supportive. 
 t: John is not responsible. 
 u: Mary is responsible. 
 ∼u: Mary isn't responsible. 
 v: John is loving and supportive. 
 ∼v: John isn't loving and supportive. | **Context:** John and Mary are parents. parents are usually loving and supportive. parents are normally responsible. Mary isn't loving and supportive. John is not responsible. 

 **Question:** Does this imply that Mary is responsible and John is loving and supportive? |
| DRD | p: Jenny and Anna are tall. 
 q: Tall people usually play basketball. 
 r: Anna is possibly an exception to this rule. 
 s: Jenny plays basketball. 
 ∼s: Jenny doesn't play basketball. | **Context:** Jenny and Anna are tall. Tall people usually play basketball. Anna is possibly an exception to this rule. 

 **Question:** Does this entail that Jenny plays basketball? |
| DRO | p: Hummingbirds are birds. 
 q: Birds migrate south for the winter. 
 r: Hummingbirds do not migrate south for the winter. 
 s: All other birds than hummingbirds migrate south for the winter. 
 ∼s: All other birds than hummingbirds don't migrate south for the winter. | **Context:** Hummingbirds are birds. Birds migrate south for the winter. Hummingbirds do not migrate south for the winter. 

 **Question:** Does this mean that all other birds than hummingbirds migrate south for the winter? |
| RE1 | p: Cats, dogs, and horses are animals. 
 q: Animals are usually considered to be intelligent creatures. 
 r: At least one of the cats or dogs is not considered intelligent. 
 s: Horses are considered to be intelligent creatures. 
 ∼s: Horses aren't considered to be intelligent creatures. 
 t: Exactly one of the cats or dogs is not considered intelligent. 
 ∼t: Exactly one of the cats or dogs is considered intelligent. | **Context:** Does this entail that horses are considered to be intelligent creatures and exactly one of cats or dogs is not considered intelligent? 

 **Question:** Does this entail that horses are considered to be intelligent creatures and exactly one of the cats or dogs is not considered intelligent? |
| RE2 | p: cats normally meow. 
 q: At least one species of cat doesn't meow. 
 r: Exactly one species of cat doesn't meow. 
 ∼r: Exactly one species of cat meows. | **Context:** Cats normally meow. at least one species of cat doesn't meow. 

 **Question:** Does this entail that exactly one species of cat doesn't meow? |
| RE3 | p: Cars have four wheels. 
 q: wheels normally have spokes. 
 r: at least one wheel does not have spokes. 
 s: Cars have four wheels with spokes. 
 ∼s: Cars don't have four wheels with spokes. | **Context:** Cars have four wheels. wheels normally have spokes. at least one wheel does not have spokes. 

 **Question:** Does this imply that cars have four wheels with spokes? |
| RAP | p: John asserts that Sally was in the store. 
 q: Jane asserts that Sally was not in the store. 
 r: John's evidence is more reliable than Jane's. 
 ∼r: John's evidence is less reliable than Jane's. 
 s: Sally was in the store. 
 ∼s: Sally wasn't in the store. | **Context:** John asserts that Sally was in the store. Jane asserts that Sally was not in the store. 

 **Question:** If John's evidence is more reliable than Jane's, does this mean that Sally was in the store? |

Table 12: Illustrative examples of NL logical context and questions created using sentences that are generated in §3.2.1 for NM logic.

| Type | Rule | FLAN-T5 | | Tk-instruct | | GPT-3 | | ChatGPT | | GPT-4 | |
|------|------|---------|---------|-------------|---------|---------|---------|---------|---------|---------|---------|
| | | $A(No)$ | $A(Yes)$ | $A(No)$ | $A(Yes)$ | $A(No)$ | $A(Yes)$ | $A(No)$ | $A(Yes)$ | $A(No)$ | $A(Yes)$ |
| PL | HS | 100 | 50.0 | 100 | 100 | 98.1 | 70.4 | 100 | 66.6 | 100 | 95.2 |
| | DS | 66.6 | 0 | 72.9 | 10.0 | 77.7 | 30.8 | 66.6 | 0 | 96.6 | 85.7 |
| | CD | 87.7 | 56.5 | 75.9 | 100 | 96.7 | 94.7 | 76.8 | 29.2 | 80.0 | 100 |
| | DD | 75.0 | 0 | 75.0 | 0 | 100 | 47.6 | 77.7 | 30.8 | 85.1 | 76.9 |
| | BD | 90.6 | 87.5 | 85.7 | 100 | 98.2 | 76.0 | 88.5 | 50 | 96.7 | 94.7 |
| | MT | 84.1 | 36.1 | 66.6 | 0 | 94.3 | 40.0 | 82.6 | 35.3 | 100 | 83.3 |
| | MI | 56.3 | 20.3 | 68.3 | 5.0 | 76.2 | 26.3 | 83.8 | 32.6 | 95.7 | 54.5 |
| | CT | 81.2 | 63.6 | 75.0 | 0 | 100 | 95.2 | 98.1 | 70.4 | 100 | 100 |
| | **Avg** | **80.2** | **39.2** | **77.4** | **39.4** | **92.6** | **60.1** | **84.3** | **39.4** | **94.3** | **86.3** |
| FOL | EG | 100 | 100 | 95.2 | 100 | 100 | 100 | 83.3 | 100 | 100 | 100 |
| | UI | 100 | 71.4 | 94.7 | 90.5 | 90.9 | 100 | 83.3 | 100 | 100 | 100 |
| | MP | 97.6 | 78.9 | 76.5 | 100 | 82.8 | 95.4 | 91.3 | 79.4 | 83.1 | 100 |
| | HS | 100 | 48.8 | 100 | 90.9 | 98.2 | 76.0 | 97.9 | 59.4 | 100 | 95.2 |
| | DS | 75.0 | 25.0 | 75.0 | 0 | 100 | 57.1 | 84.5 | 100 | 98.4 | 100 |
| | CD | 78.6 | 80.0 | 75.9 | 100 | 100 | 100 | 76.0 | 40.0 | 100 | 100 |
| | DD | 77.0 | 50.0 | 75.0 | 0 | 100 | 37.0 | 88.4 | 40.5 | 100 | 60.6 |
| | BD | 83.3 | 100 | 75.0 | 0 | 100 | 74.1 | 94.2 | 60.7 | 100 | 66.7 |
| | MT | 95.3 | 48.6 | 68.3 | 5.0 | 100 | 47.6 | 95.7 | 54.5 | 96.1 | 64.3 |
| | **Avg** | **89.6** | **67.0** | **81.7** | **54.0** | **96.9** | **76.3** | **88.3** | **70.5** | **97.5** | **87.4** |
| NM | DRI | 50.0 | 50.0 | 64.5 | 100 | 72.2 | 68.2 | 69.2 | 85.7 | 67.8 | 91.6 |
| | DRS | 67.2 | 4.5 | 74.7 | 0 | 63.0 | 0 | 74.0 | 0 | 91.6 | 38.6 |
| | DRD | 100 | 95.2 | 80.0 | 100 | 100 | 100 | 80.0 | 100 | 100 | 100 |
| | DRO | 40.0 | 40.0 | 50.0 | 0 | 100 | 90.9 | 74.1 | 100 | 68.9 | 100 |
| | RE1 | 76.9 | 28.6 | 74.7 | 0 | 97.7 | 51.3 | 80.3 | 66.6 | 100 | 55.5 |
| | RE2 | 100 | 100 | 52.6 | 100 | 100 | 100 | 55.5 | 100 | 95.2 | 100 |
| | RE3 | 72.2 | 68.2 | 79.2 | 93.8 | 69.2 | 85.7 | 62.1 | 81.8 | 86.3 | 94.4 |
| | RAP | 56.8 | 55.8 | 52.6 | 100 | 86.8 | 83.3 | 70.2 | 100 | 86.3 | 94.4 |
| | **Avg** | **70.4** | **55.3** | **66.0** | **61.7** | **86.1** | **72.4** | **70.7** | **79.3** | **87.0** | **84.3** |

Table 13: Performance of LLMs in few-shot setting in terms of label-wise accuracy on Log-icBench(Eval), where $A(Yes)$ and $A(No)$ denote the accuracy for the $Yes$ and $No$ labels, respectively.

| Type | Rules | FLAN-T5 | | Tk-instruct | | GPT-3 | | ChatGPT | | GPT-4 | |
|---|---|---|---|---|---|---|---|---|---|---|---|
| | | $A(No)$ | $A(Yes)$ | $A(No)$ | $A(Yes)$ | $A(No)$ | $A(Yes)$ | $A(No)$ | $A(Yes)$ | $A(No)$ | $A(Yes)$ |
| PL | HS | $100_{0.000}$ | $48.4_{0.006}$ | $97.9_{0.001}$ | $57.9_{0.052}$ | $97.6_{0.020}$ | $78.3_{0.083}$ | $100.00_{0.000}$ | $57.17_{0.016}$ | $100.00_{0.000}$ | $91.03_{0.041}$ |
| | DS | $64.1_{0.127}$ | $08.3_{0.144}$ | $67.9_{0.065}$ | $10.9_{0.121}$ | $75.5_{0.012}$ | $33.3_{0.577}$ | $73.77_{0.007}$ | $5.56_{0.096}$ | $100.00_{0.000}$ | $76.41_{0.072}$ |
| | CD | $50.0_{0.433}$ | $25.0_{0.000}$ | $75_{0.000}$ | $25.0_{0.000}$ | $97.7_{0.019}$ | $75.4_{0.111}$ | $99.40_{0.010}$ | $81.02_{0.053}$ | $100.00_{0.000}$ | $55.74_{0.102}$ |
| | DD | $75.0_{0.000}$ | $25.0_{0.000}$ | $75.0_{0.000}$ | $25.0_{0.000}$ | $78.0_{0.022}$ | $43.4_{0.009}$ | $100.00_{0.000}$ | $33.14_{0.029}$ | $100.00_{0.000}$ | $33.22_{0.019}$ |
| | BD | $75.0_{0.000}$ | $25.0_{0.000}$ | $75.0_{0.000}$ | $25.0_{0.000}$ | $80.5_{0.043}$ | $97.0_{0.052}$ | $97.38_{0.029}$ | $58.05_{0.035}$ | $100.00_{0.000}$ | $38.31_{0.023}$ |
| | MT | $92.2_{0.012}$ | $44.6_{0.013}$ | $74.5_{0.019}$ | $24.4_{0.030}$ | $72.5_{0.037}$ | $17.5_{0.093}$ | $92.30_{0.014}$ | $45.47_{0.020}$ | $100.00_{0.000}$ | $70.72_{0.038}$ |
| | MI | $63.6_{0.033}$ | $23.2_{0.006}$ | $64.2_{0.022}$ | $0.0_{0.000}$ | $81.5_{0.033}$ | $33.3_{0.041}$ | $91.26_{0.069}$ | $41.32_{0.054}$ | $98.81_{0.021}$ | $37.48_{0.030}$ |
| | CT | $25.0_{0.433}$ | $16.7_{0.144}$ | $78.3_{0.057}$ | $31.5_{0.112}$ | $95.8_{0.038}$ | $97.0_{0.052}$ | $100.00_{0.000}$ | $52.32_{0.034}$ | $100.00_{0.000}$ | $50.24_{0.041}$ |
| | **Avg** | $68.1_{0.130}$ | $27.0_{0.039}$ | $76.0_{0.020}$ | $25.0_{0.039}$ | $84.9_{0.035}$ | $59.4_{0.049}$ | $94.27_{0.016}$ | $46.76_{0.042}$ | $99.85_{0.003}$ | $56.65_{0.046}$ |
| FOL | EG | $100_{0.000}$ | $100_{0.000}$ | $95.0_{0.004}$ | $100_{0.000}$ | $88.9_{0.096}$ | $100_{0.000}$ | $89.72_{0.048}$ | $100.00_{0.000}$ | $100.00_{0.000}$ | $100.00_{0.000}$ |
| | UI | $98.1_{0.032}$ | $86.9_{0.038}$ | $89.3_{0.003}$ | $84.4_{0.022}$ | $88.2_{0.042}$ | $98.2_{0.030}$ | $85.11_{0.022}$ | $94.34_{0.002}$ | $93.80_{0.025}$ | $100.00_{0.000}$ |
| | MP | $99.2_{0.013}$ | $79.3_{0.003}$ | $88.6_{0.010}$ | $86.3_{0.025}$ | $81.6_{0.008}$ | $82.3_{0.057}$ | $88.48_{0.044}$ | $80.09_{0.020}$ | $80.91_{0.005}$ | $95.38_{0.045}$ |
| | HS | $100_{0.000}$ | $49.2_{0.007}$ | $100_{0.000}$ | $52.6_{0.013}$ | $94.9_{0.028}$ | $78.7_{0.057}$ | $95.75_{0.019}$ | $53.13_{0.027}$ | $99.40_{0.010}$ | $86.99_{0.068}$ |
| | DS | $72.1_{0.052}$ | $21.9_{0.132}$ | $71.4_{0.007}$ | $04.6_{0.039}$ | $81.8_{0.034}$ | $96.3_{0.064}$ | $88.20_{0.012}$ | $97.62_{0.041}$ | $91.86_{0.016}$ | $100.00_{0.000}$ |
| | CD | $75.0_{0.000}$ | $25.0_{0.000}$ | $91.7_{0.144}$ | $62.0_{0.326}$ | $93.1_{0.05}$ | $65.9_{0.102}$ | $93.69_{0.040}$ | $87.95_{0.056}$ | $100.00_{0.000}$ | $79.46_{0.078}$ |
| | DD | $75.0_{0.000}$ | $25.0_{0.000}$ | $87.4_{0.107}$ | $28.0_{0.025}$ | $75.4_{0.004}$ | $44.4_{0.509}$ | $83.86_{0.073}$ | $30.57_{0.049}$ | $100.00_{0.000}$ | $36.64_{0.017}$ |
| | BD | $25.0_{0.433}$ | $25.0_{0.000}$ | $91.7_{0.144}$ | $47.0_{0.190}$ | $77.5_{0.019}$ | $94.4_{0.096}$ | $98.75_{0.011}$ | $67.56_{0.086}$ | $100.00_{0.000}$ | $36.04_{0.024}$ |
| | MT | $93.3_{0.001}$ | $48.1_{0.016}$ | $81.8_{0.045}$ | $35.9_{0.050}$ | $74.8_{0.013}$ | $25.7_{0.103}$ | $85.86_{0.086}$ | $42.31_{0.133}$ | $96.75_{0.010}$ | $62.84_{0.047}$ |
| | **Avg** | $82.0_{0.059}$ | $51.1_{0.021}$ | $88.5_{0.051}$ | $55.6_{0.077}$ | $84.0_{0.033}$ | $76.2_{0.113}$ | $89.94_{0.039}$ | $72.62_{0.046}$ | $95.86_{0.007}$ | $77.48_{0.031}$ |
| NM | DRI | $60.4_{0.030}$ | $59.6_{0.020}$ | $52.4_{0.084}$ | $53.8_{0.081}$ | $75.0_{0.016}$ | $100_{0.000}$ | $75.63_{0.000}$ | $89.59_{0.016}$ | $69.65_{0.038}$ | $92.02_{0.010}$ |
| | DRS | $66.3_{0.006}$ | $02.9_{0.025}$ | $60.0_{0.034}$ | $03.9_{0.010}$ | $72.5_{0.012}$ | $10.1_{0.036}$ | $72.72_{0.007}$ | $0.00_{0.096}$ | $72.26_{0.025}$ | $15.72_{0.055}$ |
| | DRD | $95.0_{0.000}$ | $95.0_{0.000}$ | $88.8_{0.036}$ | $75.7_{0.124}$ | $84.7_{0.055}$ | $100_{0.000}$ | $82.22_{0.010}$ | $100.00_{0.053}$ | $77.16_{0.053}$ | $100.00_{0.000}$ |
| | DRO | $40.0_{0.061}$ | $42.6_{0.023}$ | $43.8_{0.074}$ | $45.2_{0.091}$ | $65.3_{0.033}$ | $100_{0.000}$ | $70.33_{0.000}$ | $100.00_{0.029}$ | $51.30_{0.013}$ | $66.67_{0.577}$ |
| | RE1 | $74.2_{0.027}$ | $24.2_{0.038}$ | $85.2_{0.046}$ | $28.0_{0.009}$ | $74.3_{0.006}$ | $0.0_{0.000}$ | $81.40_{0.029}$ | $33.58_{0.035}$ | $75.00_{0.000}$ | $0.00_{0.000}$ |
| | RE2 | $100_{0.000}$ | $100_{0.000}$ | $98.2_{0.030}$ | $93.8_{0.053}$ | $50.0_{0.000}$ | $0.0_{0.000}$ | $62.31_{0.014}$ | $64.73_{0.020}$ | $50.00_{0.000}$ | $0.00_{0.000}$ |
| | RE3 | $65.6_{0.042}$ | $63.0_{0.019}$ | $78.2_{0.061}$ | $57.7_{0.057}$ | $64.4_{0.010}$ | $93.6_{0.055}$ | $67.21_{0.069}$ | $82.74_{0.054}$ | $69.43_{0.045}$ | $85.56_{0.019}$ |
| | RAP | $70.1_{0.078}$ | $62.6_{0.030}$ | $76.9_{0.021}$ | $92.5_{0.014}$ | $56.8_{0.006}$ | $92.2_{0.072}$ | $58.35_{0.000}$ | $96.97_{0.034}$ | $60.04_{0.019}$ | $100.00_{0.000}$ |
| | **Avg** | $71.4_{0.031}$ | $56.2_{0.019}$ | $72.9_{0.048}$ | $56.3_{0.055}$ | $67.9_{0.017}$ | $62.0_{0.020}$ | $71.27_{0.016}$ | $70.95_{0.042}$ | $65.61_{0.024}$ | $57.50_{0.083}$ |

Table 14: Evaluation of LLMs in terms of label-wise accuracy on *LogicBench(Eval)* without chain-of-thought prompting, where $A(Yes)$ and $A(No)$ denote the accuracy for the $Yes$ and $No$ labels, respectively.

| PL | | | FOL | | | NM | | |
|---|---|---|---|---|---|---|---|---|
| **Rule** | **A(No)** | **A(Yes)** | **Rule** | **A(No)** | **A(Yes)** | **Rule** | **A(No)** | **A(Yes)** |
| HS | $88_{0.03}$ | $45_{0.35}$ | EG | $84_{0.06}$ | $79_{0.16}$ | DRI | $49_{0.03}$ | $47_{0.11}$ |
| DS | $73_{0.03}$ | $4_{0.06}$ | UI | $84_{0.1}$ | $63_{0.11}$ | DRS | $73_{0.03}$ | $26_{0.22}$ |
| CD | $80_{0.07}$ | $29_{0.04}$ | MP | $82_{0.05}$ | $72_{0.33}$ | DRD | $64_{0.02}$ | $57_{0.21}$ |
| DD | $82_{0.05}$ | $3_{0.07}$ | HS | $83_{0.06}$ | $35_{0.02}$ | DRO | $50_{0.05}$ | $49_{0.07}$ |
| BD | $76_{0.02}$ | $27_{0.06}$ | DS | $83_{0.17}$ | $1_{0.03}$ | RE1 | $79_{0.05}$ | $30_{0.05}$ |
| MT | $75_{0.02}$ | $25_{0.25}$ | CD | $72_{0.25}$ | $26_{0.02}$ | RE2 | $56_{0.18}$ | $58_{0.09}$ |
| MI | $75_{0}$ | $11_{0.19}$ | DD | $90_{0.09}$ | $25_{0.22}$ | RE3 | $50_{0.03}$ | $50_{0.04}$ |
| CT | $65_{0.07}$ | $9_{0.08}$ | BD | $86_{0.06}$ | $28_{0.03}$ | RAP | $53_{0.04}$ | $60_{0.03}$ |
| - | - | - | MT | $78_{0.06}$ | $19_{0.01}$ | - | - | - |
| **Avg** | $\mathbf{77_{0.04}}$ | $\mathbf{22_{0.14}}$ | **Avg** | $\mathbf{83_{0.1}}$ | $\mathbf{39_{0.1}}$ | **Avg** | $\mathbf{59_{0.05}}$ | $\mathbf{47_{0.1}}$ |

Table 16: Average and standard deviation across three different prompts corresponding to each inference rule for LLaMa-2-7B.

| Context | Question | Answer | Logical Steps |
|---|---|---|---|
| We know that atleast one of the following is true (1) she is not making a cake for the party and (2) he is writing a report for his boss. Note that we do not know which ones of (1) and (2) are true. It might be the case that only (1) is true, or only (2) is true or both are true. | If she is making a cake for the party, does this mean that he is writing a report for his boss? | Yes | No. 
 Logical reasoning steps: 
 1. We know that at least one of the following is true: (1) she is not making a cake for the party and (2) he is writing a report for his boss. 
 2. The question asks if she is making a cake for the party, does this mean that he is writing a report for his boss? 
 3. Since we do not know which one of the statements (1) and (2) is true, we cannot determine if she making a cake for the party means that he is writing a report for his boss. 
 4. Therefore, the answer is No. |
| We know that atleast one of the following is true (1) John is not at school and (2) Mary is taking a break from her job. Note that we do not know which ones of (1) and (2) are true. It might be the case that only (1) is true, or only (2) is true or both are true. | If John is at school, does this entail that Mary is taking a break from her job? | Yes | No 
 Logical Reasoning Steps: 
 1. We know that at least one of the following is true: (1) John is not at school and (2) Mary is taking a break from her job. 
 2. If John is at school, it means that (1) is not true. 
 3. Since (1) is not true, it means that Mary is not necessarily taking a break from her job. 
 4. Therefore, if John is at school, it does not entail that Mary is taking a break from her job. |
| We know that atleast one of the following is true (1) Scarlett is not attending tshe lecture and (2) dani is studying for her upcoming exam. Note that we do not know which ones of (1) and (2) are true. It might be the case that only (1) is true, or only (2) is true or both are true. | If Scarlett is attending the lecture, does this mean that dani is studying for her upcoming exam? | Yes | No 
 Logical reasoning steps: 
 • We know that at least one of the following is true: (1) Scarlett is not attending the lecture and (2) Dani is studying for her upcoming exam. 
 • We do not know which one of (1) and (2) is true, or if both are true. 
 • If Scarlett is attending the lecture, it means that (1) is not true. 
 • Since we do not know if (2) is true or not, we cannot conclude that Dani is studying for her upcoming exam. 
 • Therefore, we cannot say for certain if Dani is studying for her upcoming exam if Scarlett is attending the lecture. |
| We know that atleast one of the following is true (1) Chloe is studying for her exams and (2) mila is going on vacation. Note that we do not know which ones of (1) and (2) are true. It might be the case that only (1) is true, or only (2) is true or both are true. | If Chloe isn't studying for her exams, does this entail that mila is going on vacation? | Yes | No 
 Logical reasoning steps: 
 1. We know that at least one of the following is true: (1) Chloe is studying for her exams and (2) Mila is going on vacation. 
 2. If Chloe isn't studying for her exams, it means that (1) is not true. 
 3. However, we don't have enough information to determine whether (2) is true or not. 
 4. Therefore, we cannot conclude that Mila is going on vacation just because Chloe isn't studying for her exams. |
| We know that atleast one of the following is true (1) Ava is not attending the music festival and (2) she is studying for her exams. Note that we do not know which ones of (1) and (2) are true. It might be the case that only (1) is true, or only (2) is true or both are true. | If Ava is attending the music festival, does this mean that she is studying for her exams? | Yes | No. 
 Logical reasoning steps: 
 1. We know that at least one of the following is true: (1) Ava is not attending the music festival and (2) she is studying for her exams. 
 2. The question asks if Ava is attending the music festival, does this mean that she is studying for her exams? 
 3. Since we do not know which one of the statements (1) and (2) is true, we cannot conclude that if Ava is attending the music festival, she is studying for her exams. 
 4. Therefore, the answer is No. |

Table 17: Case study on generating a logical explanation for *Disjunctive Syllogism* of *propositional logic* having the worst performance by *ChatGPT* in this category

| Context | Question | Answer | Logical Steps |
|---|---|---|---|
| If someone buys a car , then they will save money on transportation . If they sell their car , then they will have more money to invest. We know that atleast one of the following is true (1) she won't save money on transportation and (2) she won't have more money to invest. Note that we do not know which ones of (1) and (2) are true. It might be the case that only (1) is true, or only (2) is true or both are true. | If at least one of (1) and (2) is true, can we say, at least one of the following must always be true? (a) sara didn't buy a car and (b) she didn't sell her car | Yes | No
Logical reasoning steps:

• If someone buys a car, they will save money on transportation.
• If someone sells their car, they will have more money to invest.
• At least one of the following is true: (1) she won't save money on transportation and (2) she won't have more money to invest.
• We do not know which one of (1) and (2) is true, or if both are true.
• The question asks if at least one of (1) and (2) is true, can we say that at least one of the following must always be true: (a) Sara didn't buy a car and (b) she didn't sell her car.
• If (1) is true, it means Sara didn't save money on transportation. This does not necessarily mean she didn't buy a car or sell her car. Therefore, we cannot say that (a) must always be true.
• If (2) is true, it means Sara didn't have more money to invest. This does not necessarily mean she didn't buy a car or sell her car. Therefore, we cannot say that (b) must always be true.
• Based on the reasoning steps, the answer is No. |
| If someone takes a walk, then they will get fresh air. If they watch tv, then they will feel relaxed. We know that atleast one of the following is true (1) Jill won't get fresh air. and (2) she won't feel relaxed. Note that we do not know which ones of (1) and (2) are true. It might be the case that only (1) is true, or only (2) is true or both are true. | If at least one of (1) and (2) is true, can we say, at least one of the following must always be true? (a) Jill doesn't take a walk. and (b) she doesn't watch tv. | Yes | No. Based on the given information, we cannot say that at least one of the following must always be true: (a) Jill doesn't take a walk, and (b) she doesn't watch TV.
Logical Reasoning Steps:
1. If someone takes a walk, then they will get fresh air.
2. If they watch TV, then they will feel relaxed.
3. We know that at least one of the following is true: (1) Jill won't get fresh air, and (2) she won't feel relaxed.
4. We do not know which one of (1) and (2) is true. It could be that only (1) is true, only (2) is true, or both are true.
5. The question asks if at least one of (1) and (2) is true, can we say that at least one of the following must always be true: (a) Jill doesn't take a walk, and (b) she doesn't watch TV. |
| If someone takes care of her health, then they will be fit and healthy. If they indulge in unhealthy habits, then they will be prone to diseases. We know that atleast one of the following is true (1) Jenny won't be fit and healthy and (2) she won't be prone to diseases. Note that we do not know which ones of (1) and (2) are true. It might be the case that only (1) is true, or only (2) is true or both are true. | If at least one of (1) and (2) is true, can we say, at least one of the following must always be true? (a) jenny doesn't take care of her health and (b) she doesn't indulge in unhealthy habits | Yes | No
Logical reasoning steps:

• From the given context, we have two conditional statements:
  – If someone takes care of her health, then they will be fit and healthy.
  – If they indulge in unhealthy habits, then they will be prone to diseases.
• We also know that at least one of the following is true:
  – Jenny won't be fit and healthy.
  – She won't be prone to diseases.
• We are asked if we can conclude whether at least one of the following must always be true:
  – Jenny doesn't take care of her health.
  – She doesn't indulge in unhealthy habits.
• Let's analyze the given options:
  If Jenny is not fit and healthy [(1) is true], according to the first conditional statement, she might not be taking care of her health. So, (a) could be true.
  If Jenny is not prone to diseases [(2) is true], according to the second conditional statement, she might not be indulging in unhealthy habits. So, (b) could be true.
• Based on the information provided in the context and the conditional statements, we cannot definitively conclude that at least one of the options (a) or (b) must always be true. It depends on the specific reasons for Jenny not being fit and healthy or not being prone to diseases. |
| If someone reads a lot, then they will get good grades. If they watch television, then they will have a sound sleep. We know that atleast one of the following is true (1) Jill won't get good grades and (2) she won't have a sound sleep. Note that we do not know which ones of (1) and (2) are true. It might be the case that only (1) is true, or only (2) is true or both are true. | If at least one of (1) and (2) is true, can we say, at least one of the following must always be true? (a) Jill doesn't read a lot and (b) she doesn't watch television | Yes | No, we cannot say that at least one of the following must always be true: (a) Jill doesn't read a lot, and (b) she doesn't watch television.

Logical Reasoning:
1. We know that if someone reads a lot, then they will get good grades.
2. We also know that if someone watches television, then they will have a sound sleep.
3. We are given that at least one of the following is true: (1) Jill won't get good grades and (2) she won't have a sound sleep.
4. It is possible that only (1) is true, meaning Jill won't get good grades but she will have a sound sleep.
5. It is also possible that only (2) is true, meaning Jill will get good grades but she won't have a sound sleep.
6. It is also possible that both (1) and (2) are true, meaning Jill won't get good grades and she won't have a sound sleep.
7. If only (1) is true, then it means Jill doesn't read a lot, but we don't know if she watches television or not.
8. If only (2) is true, then it means Jill doesn't watch television, but we don't know if she reads a lot or not.
9. If both (1) and (2) are true, then it means Jill doesn't read a lot and she doesn't watch television.
10. Based on the given information, we cannot say for certain whether at least one of the following must always be true: (a) Jill doesn't read a lot and (b) she doesn't watch television. |
| If someone takes a test, then they will get a good grade. If they do not revise enough, then they will not score well. We know that atleast one of the following is true (1) John won't get a good grade and (2) he will score well. Note that we do not know which ones of (1) and (2) are true. It might be the case that only (1) is true, or only (2) is true or both are true. | If at least one of (1) and (2) is true, can we say, at least one of the following must always be true? (a) john doesn't take a test and (b) he revises enough | Yes | No
Logical reasoning steps:

1. We know that if someone takes a test, they will get a good grade, which can be represented as: Test → Good Grade.
2. We also know that if someone does not revise enough, they will not score well, which can be represented as: Not Enough Revision → Not Score Well.
3. We are given that at least one of the following is true: (1) John won't get a good grade and (2) he will score well.
4. This means that either Test → Good Grade is true (2 is true) or Not Enough Revision → Not Score Well is true (1 is true), or both are true.
5. Now, let's analyze the implications of (a) and (b):
  (a) John doesn't take a test: This statement is not directly provided in the context, so we cannot definitively say whether John takes the test or not based on the given information.
  (b) He revises enough: This statement is also not directly provided in the context, so we cannot definitively say whether John revises enough based on the given information.
6. In conclusion, based on the information provided in the context, we cannot determine whether (a) John doesn't take a test or (b) he revises enough. Therefore, we cannot say that at least one of these statements must always be true. |

Table 18: Case study on generating a logical explanation for *Destructive Dilemma* of *first order logic* having the worst performance by *ChatGPT* in this category

| Context | Question | Answer | Logical Steps |
|---------|----------|--------|---------------|
| John and Mary are parents. parents are usually loving and supportive. parents are normally responsible. Mary isn't loving and supportive. John is not responsible. | Does this imply that Mary is responsible and John is loving and supportive? | Yes | No
Logical Reasoning Steps:
1. The context states that parents are usually loving and supportive.
2. The context also states that parents are normally responsible.
3. The context further states that Mary isn't loving and supportive.
4. The context states that John is not responsible.
5. Based on the context, it can be inferred that Mary is not responsible because she is not loving and supportive.
6. It can also be inferred that John is not loving and supportive because he is not responsible.
7. Therefore, the context does not imply that Mary is responsible and John is loving and supportive. |
| apples and bananas are fruits. fruits are usually sweet. fruits are normally juicy. apples are not sweet. bananas are not juicy. | Does this mean that bananas are sweet and apples are juicy? | Yes | No.
Logical Reasoning Steps:
1. According to the context, apples and bananas are fruits.
2. The context also states that fruits are usually sweet and normally juicy.
3. However, it is mentioned that apples are not sweet and bananas are not juicy.
4. Therefore, based on the given information, we can conclude that bananas are not sweet and apples are not juicy.
5. The question asks if bananas are sweet and apples are juicy.
6. Based on the reasoning steps, the answer is No, bananas are not sweet and apples are not juicy. |
| cats and dogs are animals. animals have four legs. animals have fur. cats have no fur. dogs are not four-legged. | Does this mean that dogs have fur and cats have four legs? | Yes | No.
Logical reasoning steps:
1. Cats and dogs are animals.
2. Animals have four legs.
3. Animals have fur.
4. Cats have no fur.
5. Dogs are not four-legged. |
| birds and fish are animals. animals have four legs. animals are normally omnivores. birds don't have four legs. fish is not an omnivore. | Does this imply that fish have four legs and birds are omnivores? | Yes | No
Logical Reasoning Steps:
1. The context states that animals have four legs.
2. Birds are animals, but they don't have four legs.
3. Therefore, it can be concluded that not all animals have four legs.
4. The context also states that animals are normally omnivores.
5. Fish is an animal, but it is not an omnivore.
6. Therefore, it can be concluded that not all animals are omnivores.
7. Based on the reasoning steps, it cannot be implied that fish have four legs and birds are omnivores. |
| Blocks A and B are heavy. Heavy blocks are normally located on the table. Heavy blocks are normally red. A is not on the table. B is not red. | Does the context imply that B is on the table and A is red in color? | Yes | No, the context does not imply that B is on the table and A is red in color.

Logical reasoning steps:
1. The context states that heavy blocks are normally located on the table.
2. It also states that A is not on the table.
3. Therefore, we can conclude that A is not located on the table.
4. The context also states that heavy blocks are normally red.
5. It further states that B is not red.
6. Therefore, we can conclude that B is not a heavy block.
7. Since B is not a heavy block, it does not necessarily need to be located on the table.
8. Additionally, the context does not provide any information about the color of A.
9. Therefore, we cannot conclude that A is red in color.
10. In conclusion, the context does not imply that B is on the table and A is red in color. |

Table 19: Case study on generating a logical explanation for *Default Reasoning with Several Default* of *Non-monotonic logic* having the worst performance by *ChatGPT* in this category

| Inference rules | Contxet | Story |
|---|---|---|
| HS | If Jerry goes for jogging, then he will work up a sweat. If he will work up a sweat, then he will drink lots of water. | Jerry was determined to stay healthy and fit so he made sure to go for jogging every day. Sure enough, after his jogs he would work up a sweat and, in order to stay hydrated, would drink lots of water. |
| DS | We know that atleast one of the following is true (1) Ava is not attending the music festival and (2) she is studying for her exams. Note that we do not know which ones of (1) and (2) are true. It might be the case that only (1) is true, or only (2) is true or both are true. | Ava had an important decision to make - attend the music festival or study for her exams. She couldn't do both, so she had to choose. In the end, she decided that either she would attend the music festival or she would study for her exams - or both, but only one at a time. She wasn't sure which one she wanted to do, but she was determined to do one of them. |
| CD | If Lisa eats breakfast in the morning, then she will have energy to start my day. If she sleeps in late, then she will feel groggy in the morning. We know that atleast one of the following is true (1) Lisa eats breakfast in the morning and (2) she sleeps in late. Note that we do not know which ones of (1) and (2) are true. It might be the case that only (1) is true, or only (2) is true or both are true. | Lisa always faces a tough challenge every morning: eat breakfast or sleep in late. On one hand, if she eats breakfast in the morning, she will have enough energy to get her day started. On the other hand, if she sleeps in late, she will feel groggy in the morning. Despite Lisa's dilemma, one thing is for certain - either she eats breakfast or she sleeps in late. |

Table 20: Examples of stories generated from original context for three distinct propositional rules. HS: Hypothetical Syllogism, DS: Disjunctive Syllogism, CD: Constructive Dilemma

| Inference Rules | Original | | Story-like | |
|---|---|---|---|---|
| | $A(Yes)$ | $A(No)$ | $A(Yes)$ | $A(No)$ |
| HS | 52.63 | 100 | 73.91 | 94.74 |
| DS | 50.00 | 81.25 | 16.67 | 72.58 |
| CD | 39.58 | 96.88 | 45.16 | 87.76 |
| DD | 33.33 | 87.50 | 24.00 | 74.55 |
| BD | 39.58 | 96.88 | 41.03 | 90.24 |
| MT | 45.45 | 89.36 | 0 | 61.54 |
| MI | 25.68 | 83.33 | 34.21 | 83.33 |
| CT | 45.45 | 100 | 46.15 | 95.12 |
| **Average** | **41.46** | **91.90** | **35.14** | **82.48** |

Table 21: Performance of GPT-3 given *Prompt 1*(§H.2) in terms of label-wise accuracy on PL with original and story-like context from *LogicBench(Eval)*.

