# OpenReview forum: "LogicBench: Towards Systematic Evaluation of Logical Reasoning Ability of Large Language Models"
_ICLR.cc/2024/Conference — Submitted to ICLR 2024_

### Official Review · Reviewer_Z2AS · 2023-10-30

**Soundness:** 4 excellent
**Presentation:** 3 good
**Contribution:** 3 good
**Rating:** 6
**Confidence:** 5

**Summary:**

The authors present LogicBench, a natural language QA dataset covering 25 combinations of inference rules (e.g., modus ponens, disjuntive syllogism, etc.) and logics (propositional, first-order predicate, and non-monotonic). The generation procedure is clearly described and several examples are shown. A number of models are evaluated on the benchmark, and it is also used as a fine-tuning dataset to assess resultant improvements on other logical reasoning benchmarks. Related work is compared to and discussed throughout.

**Strengths:**

This is a very nicely designed benchmark that I am excited to use. I appreciate the effort the authors have taken on a number of fronts. 1) the authors unify a number of logics and inference rules into one global benchmark. 2) the materials are synthetically generated, but still reasonably naturalistic. 3) the paper is clear and well-written. 4) the contextualization with respect to other benchmarks breaks the space down very elegantly.

This paper is already marginally above the acceptance threshold and can be bumped to an accept if the below questions are addressed.

**Weaknesses:**

There are several opportunities to improve the contribution and presentation of the work. I will address these more specifically below under questions. 1) There are a number of places where the reader would appreciate more detail as to the background behind decision points during the creation of this benchmark. 2) Aside from inference rules, there are additional correlated metadata that would be helpful to add and consider when comparing model performance across subsets. 3) it is not clear what the reader gains from the fine-tuning experiment.

**Questions:**

- Since the synthetic generation process is completed partially via sampling from an LLM, there are of course concerns as to whether the generated problems actually follow the intended logical structure and are devoid of other errors. The authors dutifully note this, and as such, human-validate a subset of the data (500 instances), which they dub LogicBench[Eval] as opposed to the larger set (3750 instances) that are not validated in LogicBench[Aug]. It is perfectly reasonable to only validate a subset of the data given the size, but the authors should include a breakdown of the types of errors found when manually validating this subset. A table should be added describing sample statistics e.g., for a sample of 100 problems generated (as might be found in the raw Aug portion), how many need to be corrected for various discrepancies in logical formulation, typos, grammar, etc. This will help infer the ceiling on the Aug dataset.
- In the description of the NL conversion, the authors note the following:
`For instance, implication: “p → q” is
expressed as “If p, then q”, conjunction: “p ∧ q” is expressed as “p and q.”, and disjunction: “p ∨ q”
is expressed as “At least one of the following is true: (1) p and (2) q. Note that we do not know which
of (1) and (2) is true. It is possible that only (1) is true, or only (2) is true, or both are true.”`
Can the authors comment on the unwieldy specification of "or"? I understand that the authors are attempting to explicitly distinguish "or" from "xor", but should make this clear and expand on why simpler versions were insufficient. Was a simpler specification attempted initially but resulted in extremely low performance across the board due to disambiguation difficulties? More information is needed here.
- Some clarifying information on the metric is required. If I understand correctly, each instance has 4 variants? 1/4 of these shows the correct application of the rule, later measured by A(Yes), and 3/4 show variants that cannot satisfy the conclusion, later measured by A(No)? If this is not correct, perhaps this section can be clarified for other readers as well. For the A(No) problems, it appears to me that these contain both examples where the evidence provided cannot prove the conclusion or its negation, but also instances, where the evidence provided, can in fact prove the negation of the conclusion. For example, in Figure 2 Stage 3, Variation 2 implies "not S1", whereas Variations 3 and 4 can neither prove "S1" nor "not S1". Perhaps these patterns of variants should be noted differently in the dataset metadata. I would suspect certain audiences to care about the distinction between these types of errors.
- Finally, it is unclear to me what we gain from the fine-tuning experiments. The performance increases on other benchmarks are minimal and it seems to me space would be better spent clarifying the benchmark details, such as those I've raised above, as these reflect the core contribution of the paper. I would suggest moving this to the appendix. If other reviewers disagree with me and find them informative, this suggestion may be disregarded and instead, the contribution explained to the reader more explicitly.

---

> ### Author Response · Authors · 2023-11-21
>
> R1: Clarification on why simpler versions were insufficient for ‘or’.
>
> A1: Thank you for this comment. Yes, indeed we want to make an explicit distinction between ‘or’ and ‘xor’. We found that understanding the logical implication of 'or' when integrated into logical formulations posed challenges to both humans - our human evaluators, mostly graduate students - and models in terms of understanding it. So with the focus of the paper on the inference rules, we decided to make the desired meaning of `or’ explicit using the above construct, so that our human evaluations and annotations would be correct. See the below example from LogicBench(Aug) where the use of our template vs. the direct 'or' formulation:
>
> ```
> Context: If Sam has not finished his homework, then he will not go to the party.
>
> Contextual Template Question (from our dataset): "Based on the context, can we ascertain that at least one of the following must always be true: (a) Sam has finished his homework, and (b) He will not attend the party?"
>
> Direct 'or' Question: "Can we conclude that Sam has finished his homework or he will not attend the party?"
> ```
>
> From the example provided above, it becomes evident that a question using a direct 'or' can be challenging to interpret and may cause confusion. Hence, we choose to go with the more expressed formulation of ‘or’.
>
> ----
> R2: Additional information on the metric and about meta-data
>
> A2: Yes, your understanding is indeed correct. The question with the answer ‘yes’ has the correct application of the rule, and the question with the answer ‘no’ has a conclusion that is not supported by context (rule). Furthermore, we agree that different variation types can be added as metadata for a better understanding of data. Thanks for this valuable suggestion, and we will incorporate it in the revised version.
>
> -----
> R3: Details Related to fine-tuning experiments presented in Sec. 4.3
>
> A3: Thank you for this suggestion. With our exploration of augmenting data for training and showing that it is useful for generalizing to real-world logic datasets, we demonstrate an additional use case of LogicBench. Thus, we consider these experiments valuable. However, the core contribution remains to be systematical evaluation and highlighting flaws in the logical reasoning ability of LLMs.
>
> -----
> Note: During the rebuttal period, we conducted a human evaluation on a subset of LogicBench(Eval). We will share the results earliest possible.

---

> ### Author Response · Authors · 2023-11-22
>
> R4: Small-scale Human Evaluation on LogicBench(Eval)
>
> A4: Thank you for this suggestion. Please note that, in Sec. 3.3, we performed a systematic evaluation to ensure that generated instances follow the intended logical structure. To further support the integrity and reliability of the benchmark, we hired three graduate student volunteers to manually check the quality of generated data instances. Due to the limited rebuttal period, we randomly selected 5 instances from each inference rule, resulting in a total of 125 instances across 25 reasoning patterns for human evaluation. In particular, annotators are asked to provide binary answers (yes/no) for “Validity of generated context”, and “Validity of generated question” to make sure they adhere to the intended logical structure. Each instance is annotated by three different annotators. The inter-annotator agreement, measured with raw/observed agreement is 0.956. When there was a disagreement between annotators, a majority class was preferred. We provided example task instructions at https://anonymous.4open.science/r/LogicBench-EEBB/human_eval/readme.md. Results reveal that, after majority voting, annotators provided ‘yes’ for all 125 instances showing that all given instances adhere to the intended logical structure which further supports the quality of data checked by authors. We added this study in Appendix L.

---

### Official Review · Reviewer_8r5S · 2023-11-02

**Soundness:** 3 good
**Presentation:** 3 good
**Contribution:** 2 fair
**Rating:** 5
**Confidence:** 2

**Summary:**

This work proposes a logical reasoning benchmark LogicBench that evaluates the logical reasoning abilities of large language models. LogicBench includes binary Yes/No questions of several propositional logic rules, first-order logic extended from the propositional logic rules and non-monotonic reasoning. The sentence pairs used to generate the dataset instances are generated by GPT-3. The authors observed that current models like FLAN-T5, Tk-instruct, and GPT series have better performance on the negative labels (No) than the positive labels (Yes). Further analysis demonstrates that ChatGPT has challenges in fully comprehend many reasoning problems and factors like negations and rule complexity bring challenges in logical understanding.

**Strengths:**

- The logical reasoning problem is an important research direction in revealing the reasoning abilities of current pretrained language models. LogicBench includes several logical categories to picture the logical reasoning ability.
- The analysis about unfolding the reasoning steps and investigating influential factors is helpful in diagnose the bottleneck of reasoning abilities.

**Weaknesses:**

- In general, it is not easy to conclude the findings of the evaluation results from LogicBench and how it actually demonstrates the logical reasoning abilities. For example, whether the prompting method is the best approach to let the model speak out the reasoning steps that corresponds to the reasoning decision given the variances and limitations of generations of GPT-3/4. Similarly, other discussions need more evidences to provide enough information to support the suspensions.

**Questions:**

- What are the source of the sentences used to generate the pairs in Figure 2 (e.g. 'Liam finishes his work' in Table 4)?
- Are Yes and No questions balanced in the benchmark?
- The authors indicate that the LogicQA and ReClor include richer reasoning formats, but will it necessarily undermine the impacts of finetuning on LogicBench?

---

> ### Author Response · Authors · 2023-11-21
>
> R1: Justification on why we use the prompting method to evaluate LLMs’ reasoning ability
>
> A1: We note that the emergence of the prompt paradigm has shifted the assessment of LLMs, favoring zero-shot and few-shot evaluations over traditional supervised fine-tuning (SFT). Thus, our work also adapts a similar method and evaluates LLMs using Zero-shot-CoT, a standard practice for meaningful reasoning with LLMs. While evaluating LLMs such as GPT-3/4, we set the temperature hyper-parameter to 0 to control the variation in generation. Furthermore, we also conducted a manual qualitative analysis (section 4.3) to support our findings.
>
> ---
> R2: What are the source of the sentences used to generate the pairs in Figure 2?
>
> A2: We prompt the model (GPT-3) to generate source sentences (propositions) as described in Sec 3.2.1. The prompt template is presented in Figure 3, and the comprehensive prompt is presented in Appendix A.
>
> ----
> R3: Are Yes and No questions balanced in the benchmark?
>
> A3: No, they are not balanced. Thus, we presented a separate accuracy in Table 6, A(Yes) and A(No) to evaluate model performance. We updated this in the paper (Section 3.3).
>
> -----
> R4: The authors indicate that the LogicQA and ReClor include richer reasoning formats, but will it necessarily undermine the impacts of finetuning on LogicBench?
>
> A4: LogicBench is designed to evaluate LLMs on mainly single-step reasoning across a range of inference rules and show that they struggle with even single-step reasoning for complex inference rules. Furthermore, errors made during the initial single-step reasoning can be propagated through subsequent reasoning steps, potentially leading to wrong conclusions in multi-step reasoning scenarios. Thus, teaching models first to do better single-step reasoning via different approaches (fine-tuning in our case) and then utilizing them to perform reasoning over complex and longer formats (LogiQA, ReClor) can improve their performance (as shown in Section 4.3). Hence, we believe that indicating LogicQA and ReClor include richer reasoning formats will not necessarily undermine the impacts of finetuning on LogicBench.

---

### Official Review · Reviewer_sePi · 2023-11-04

**Soundness:** 3 good
**Presentation:** 3 good
**Contribution:** 3 good
**Rating:** 5
**Confidence:** 2

**Summary:**

This paper proposes a new benchmark for the evaluation of the logical reasoning ability of large language models. The main claim is that this new benchmark comparatively considers new types of reasoning including non-monotonic reasoning that is not considered in the existing ones. They evaluate multiple language models on this benchmark and provide further analysis of the difficulties those models have in reasoning. Finally, they show their synthesized data help models in improving logical reasoning when it is used in their LLM training (here T5) and tested on existing logical reasoning benchmarks.

**Strengths:**

They created a new benchmark/resource for QA when complex logical reasoning is needed to answer the questions.
Compared to the existing works, this paper is more extensive in its investigation of the types and patterns of logical reasoning needed for QA. They made sure to consider all different kinds including 25 inference rules.  The dataset can serve as a source of both evaluation and fine-tuning. The results are more detailed with respect to the difficulty of various reasoning rules and examined for a variety of language models.

**Weaknesses:**

—The work is incremental in extending the existing benchmarks and providing more results of LLM evaluations on logical reasoning.
—Sometimes, I am not sure if the provided conclusions are reliable in general.  For example, being more correct in NO answers might be due to having bias to saying more NOs than Yes’s and not more than that.
Intuitively, PL should be easier than FOL. The analysis does not bring any insight into why this is not the case with LLMs reasoning. Again the conclusion must be affected by the biases in the data creation. Often, making general conclusions is hard for this synthesized benchmarking practice and from that perspective, this work is not progressive and remains under the drawbacks of similar previous work.

**Questions:**

N/A

---

> ### Author Response · Authors · 2023-11-21
>
> R1: “being more correct in NO answers might be due to having bias to saying more NOs than Yes’s and not more than that”
>
> A1: Please note that we presented our results separately for A(No) and A(Yes). Moreover, we used CoT to evaluate the model which is the standard practice for meaningful reasoning with LLMs, and also make sure that the final answer is based on logical reasoning steps but not simple heuristics/shortcuts. Hence, we believe that incorrect/correct answers (Yes/No) produced by LLMs are not simply because of bias but because of the underlying reasoning process. To further support this, we also conducted a manual qualitative analysis (section 4.3) to present our findings.
>
> ----
> R2: Justification of lower performance of GPT-4 on PL as compared to FOL
>
> A2: Thank you for raising this point. Please note that the results in Table 6 show an average improvement in FOL results because of LLMs’ high accuracy on two axioms, EI and UI. However, when we compare the performance across the six common inference rules between PL and FOL (MT, HS, DS, BD, CD, DD), GPT-4 achieves an average of 49.16% A(Yes) for PL and 39.54% A(Yes) for FOL which shows that our results show the expected behavior.
>
> The high accuracy of GPT-4 in handling EI and UI can be attributed to their simplicity.  While from human experience and complexity theory, FOL is harder than PL in general; in the LLM context, the crucial factor becomes what kind of logical sentences LLMs are pre-trained on. It seems that LLMs are pre-trained more on simple FOL sentences than on simple PL sentences. To get an indication of this we gave GPT-4 prompt_1 - “Give twenty statements that have ‘if’ and ‘then’ in them” and prompt_2 - “Give twenty statements that have material implication "or" in them”. From the results, we can observe that 11 of the 20 sentences in response to the first prompt were FOL kind and only 9 were propositional kind. On the other hand, all 20 statements in the response to the second prompt were FOL sentences. This shows that LLMs’ comprehend simple FOL sentences, thus, showing high accuracy on simpler FOL axioms, i.e., EI and UI. However, LLMs exhibit consistently low performance across other complex/longer inference rules that are common to both PL and FOL (e.g., CD, BD, etc.).

---

> ### Comment · Reviewer_sePi · 2023-11-22
>
> Thanks for the author's response, I read the responses and the new analysis provided in the rebuttal phase. I think the paper is a good one as mentioned in my initial review.

---

### Official Review · Reviewer_1d9G · 2023-11-06

**Soundness:** 3 good
**Presentation:** 3 good
**Contribution:** 3 good
**Rating:** 6
**Confidence:** 5

**Summary:**

This paper contributes Logic-Bench, a diagnostic dataset for systematic evaluation of logical reasoning capability of large language models. It covers 25 different reasoning patterns including propositional, first-order, and non-monotonic logics. The paper presents an evaluation for different LLMs in both zero-shot and few-shot settings. The authors also showed that LLMs trained with Logic-Bench yield better performance on several logical reasoning datasets such as LogiQA and ReClor.

**Strengths:**

- The paper contributed a useful diagnostic dataset for evaluating the logical reasoning capability of large language models.
- It covers a diverse range of logic reasoning types, especially the under-explored non-monotonic logics, which would be beneficial for future research.
- The evaluation on different LLMs are comprehensive.
- The paper is well-written and easy to follow.

**Weaknesses:**

1) The main concern is that the dataset is synthetically generated with pre-defined logical rules and templates. Although this gives a comprehensive categorization of the logic types, it also leaves the question of how well it can generalize to real-world logical reasoning tasks? Although in Table 7 the authors showed that fine-tuning T5 with LogicBench leads to better performance, the performance improvement is quite marginal.
2) A salient drawback is the synthetic nature of the dataset, crafted using predetermined logical rules and templates. This raises the question of its applicability and relevance to real-world logical reasoning challenges. While the authors showed in Table 7 that fine-tuning T5 with LogicBench leads to better performance, this enhancement was relatively slight.
3) The few-shot learning performance from Table 13 hint at the potential of LogicBench to soon become less challenging. Remarkably, with few-shot prompting, GPT-4 already achieves an accuracy of roughly 90% in PL and FOL and about 85% for NM.
4) The paper lacks the evaluation on modern open-sourced LLMs such as LLAMA, Alpaca, Vicuna, etc.
5) Since the dataset is quite synthetic, I assume that when fine-tuning the model on a few examples, the model can achieve quite good performance. It would be beneficial for the authors to report this supervised performance as a reference for the potential upper-bound performance.

**Questions:**

- Given the first weakness, could the authors elaborate on:
  1. Whether high/low performance on LogicBench truly indicates high/low real-world logical reasoning capabilities for LLMs?
  2. Is there evidence to support the premise that training LLMs on synthetic logic data such as those in LogicBench genuinely enhances their general logical reasoning capabilities?
- In addition to reporting the break-down performance of each reasoning type, it would be intriguing to see if the model has the capability for cross-type generalization. For instance, when trained exclusively on one reasoning form (like PL), can the model generalize effectively to another, such as FOL?

---

> ### Author Response · Authors · 2023-11-21
>
> R1: Discussion on how well LogicBench can generalize to real-world logical reasoning tasks
>
> A1: Thanks for this comment. Please note that the core contribution of our proposed work is the diagnostic benchmark for systematically evaluating the logical reasoning ability of LLM across 25 different reasoning inference rules. This benchmark goes beyond simple inference rules in classical logic and incorporates more complex inference rules including NM reasoning. Moreover, the evaluation of various LLMs on LogicBench(Eval) leads to interesting findings and shows shortcomings in their ability to accurately solve even single-step reasoning problems. Furthermore, with our exploration of augmenting data for training and showing that it is useful for generalizing to real-world logic datasets, we demonstrate an additional use case of LogicBench. However, the core contribution remains to be systematical evaluation and highlighting flaws in the logical reasoning ability of LLMs.
>
> ----
> R2: Justification of high few-shot performance of GPT-4 on LogicBench
>
> A2: In the few-shot setting, we provide exemplars corresponding to the inference rule required to answer the test question. Bigger models such as GPT-4 have been shown to be remarkably good at following the in-context exemplars and mimicking the process to reach the correct conclusions. Thus, leveraging the in-context exemplars, GPT-4 achieves high accuracy in a few-shot setting (Table 13). In contrast, the zero-shot setting evaluates the true capabilities of these models in carrying out logical reasoning, as we can not expect that oracle exemplars corresponding to particular logical inference rules will always be available as part of prompts. We also believe that something as fundamental as logical reasoning ability should be an inherent capability of the model which is evaluated in a zero-shot setting. We discuss this point in Section 4.1.
>
> ----
> R3: Including results of open-source LLaMa-2-7B on LogicBench
>
> A3: Thanks for this suggestion. We added an evaluation of LLaMa-2-7B on LogicBench(Eval) in Appendix G. Similar to the other LLMs such as GPT-3/4 in zero-shot setting, LLaMa-2 does not fair well on LogicBench(Eval).
>
> ----
> R4: Supervised fine-tuning results on LogicBench using T5-large
>
> A4: As rightly mentioned by the reviewer, LogicT5 indeed achieves high accuracy (∼97%) on LogicBench(Eval).
>
> ----
> R5: Whether high/low performance on LogicBench truly indicate high/low real-world logical reasoning capabilities for LLMs?
>
> A5: While we agree that real-world logical reasoning can be more complex and require multi-step reasoning, we would like to emphasize that our benchmark represents the first step in evaluating LLM across a spectrum of logical inference rules and shows that these LLMs struggle even with the single-step reasoning process when logical rules become complex.  Furthermore, errors made during the initial single-step reasoning can propagate through subsequent reasoning steps, potentially leading to wrong conclusions in multi-step reasoning scenarios (which is more human-like reasoning). While our benchmark primarily addresses single-step reasoning and doesn't specifically delve into multi-step human-like reasoning, evaluation on LogicBench is a strong indicator of real-world logical reasoning capabilities for LLMs.
>
> ----
> R6: Is there evidence to support the premise that training LLMs on synthetic logic data such as those in LogicBench genuinely enhances their general logical reasoning capabilities?
>
> A6: Some recent works such as [1] show that training on synthetic data can improve the performance of out-of-domain logical reasoning datasets indicating enhanced general logical reasoning capabilities.
>
> [1] https://arxiv.org/pdf/2310.00836.pdf
>
> -----
> R7: Discussion on LLMs’ capability for cross-type generalization
>
> A7: Thanks for this valuable suggestion. During the rebuttal phase, we conducted suggested experiments and included results in Appendix K in the revised version. Experimental results reveal that the model fine-tuned on PL performs fairly well on FOL (~ 86%), though it remains lower than the supervised model fine-tuned on FOL (~ 97%). A similar observation is made for the model fine-tuned on FOL which fairly does well on PL. However, both these models struggle to generalize on non-classical NM reasoning showing lower accuracy (~ 52%). Additionally, the model fine-tuned on NM reasoning struggles in generalizing to classical logic, PL, and FOL.

---

### Official Review · Reviewer_5vLY · 2023-11-07

**Soundness:** 1 poor
**Presentation:** 3 good
**Contribution:** 2 fair
**Rating:** 5
**Confidence:** 4

**Summary:**

This paper introduces a new benchmark called LogicBench into the field of LLM reasoning. It consists of synthetically generated examples in natural language that follow a set of templated formulas derived from 3 different branches of deductive logic (propositional/first-order/non-monotonic). The examples are categorized into the various inference rules they are derived from and there is a 3 step generation procedure described. 2 sets of experiments are done: Evaluations on recent strong LM's are used to claim that LM's perform poorly on many of these structures, and the further evidence of the datasets utility is to show that it can be used as a training set for improving performance of downstream reasoning tasks.

**Strengths:**

First of all, there is a sore need for better benchmarks that can help research in advanced LLM reasoning progress further. This dataset is aimed at that need, and in at least one sense it makes a useful contribution: it adds more complex reasoning schemas that  previously have not been used, especially on the non-monotonic side.  Non-monotonic reasoning is severely understudied in LLM's but it is a crucial part of human-like reasoning, so we will need more resources on this topic.

The rigorous characterization of the reasoning types by the inference rules used can be helpful in downstream uses and analysis. I also appreciate the use of characterizations of NM taken from the classical literature which have been under-used by modern NLP.

The variation generation part of the generation pipeline is a good idea and helps make the evaluation more robust.

The extra set of experiments showing that LogicBench(Aug) can be helpful in multi-task training of logic tasks is a useful conclusion.

Performance on some of these categories seem to go down with larger model, which if they hold up (see my concern below) would make them a good candidate for Inverse Scaling [1].


[1] https://arxiv.org/abs/2306.09479

**Weaknesses:**

Notwithstanding my #1 strength above, I have to say that I have mixed feelings about the overall utility of this work to reasoning. In a sense, LogicBench is a sort of advanced version of ProofWriter type datasets, and so while I think it will have some marginal utility in the field. it also takes it in the wrong direction in a sense (in my opinion) by focusing on a set of clearly delineated but complex and unusual inference steps, whereas advanced "human-like" reasoning for LLMs will require doing multiple steps of reasoning in a very complex and realistic context where there are multiple sources of ambiguity to contend with. Examples derived from templated inference rules don't solve this problem.

My biggest concern is that the results of GPT-4 on even the simpler PL problems don't even beat a majority baseline of 50%, which seems very unexpected and possibly a bug of some kind. The authors do not even mention this surprising finding, which makes me wonder if I have misunderstood the experiments (Each task is binary Y/N, so random should be 50% ?)

With a complex benchmark like this, one generally hopes that there have been systematic checks for the integrity and reliability of the benchmark as having sensible answers and tracking the phenomenon to be measured. The authors only offer a vague assurance in the end of sec 3.3 without concrete evidence; i would love to have seen something like a small-scale human eval  on a subset to confirm the validity of the benchmark.

Overall the paper is fairly clear in what it is doing, but there are places where the authors are vague, esp. sec 4.3 first section. Most of the points in this section aren't big surprises.

**Questions:**

1. You say this is the first benchmark for NMR, but there have been others e.g. BoardGameQA [1]

2.  Table 2, i think the definition of EI is backwards.

3. It seems when we generate natural language instantiations of each rule, we expect 'if' to always be treated as material implication, but it is well known that depending on the context 'if' can be interpreted in many other ways e.g. as a subjunctive, so it may be the case that the  more natural interpretation of a particular context sentence generated by the pipeline is such. Is this a problem for the reliability of the benchmark?

4. Another surprise for me is that the PL benchmarks do worse with GPT-4 than NM, which again defies expectations. How do authors explain this?

5. Is LogicT5 pre-trained from scratch on LogicBench(Aug) using the T5 model recipe or do you train on LogicBench starting from a pretrained T5-Large checkpoint?

6. sec 3.2.1: the purpose of this step is to generate single atomic propositions. it seems we do this by generating entire triples for particular inference steps (like Modus Tollens) and then only extracting the individual atomic propositions. Why this complicated process?


[1] https://arxiv.org/abs/2306.07934

---

> ### Author Response · Authors · 2023-11-21
>
> R1: Further Discussion on the role of LogicBench towards “human-like” multiple steps reasoning
>
> A1:  We note that our benchmark represents a crucial first step in evaluating a variety of LLMs across a spectrum of logical inference rules and shows that these LLMs struggle even with the single-step reasoning process when logical rules become complex.  Furthermore, errors made during the initial single-step reasoning can be propagated through subsequent reasoning steps, potentially leading to wrong conclusions in multi-step reasoning scenarios (which is more human-like reasoning). While our benchmark primarily addresses single-step reasoning and doesn't specifically delve into multi-step human-like reasoning, it leads to valuable findings that LLMs struggle with even single-step reasoning for complex inference rules. Moreover, even in inference rules (e.g., HS, CD, DD, etc.), multiple sub-steps are involved in matching with the appropriate parts of the inference rule.
>
> --------
> R2: Clarification on task and further justification of GPT-4 results on simpler PL problems
>
> A2: Each task is a binary classification task with a Yes/No answer. Note that, the results in Table 6 show that GPT-4 demonstrates superior performance for simpler PL inference rules such as HS and MT as compared to others in PL which is expected. However, as we introduce more complex PL inference rules (i.e., longer inference rules that are not commonly used in human writing) such as DD, BD, and CD, GPT-4 starts to show a drop in performance. This is one of the findings in the paper as discussed in Section 4.3 (“Longer inference rules are still challenging”).
>
> ------
> R3: Additional details on findings that are presented in Sec 4.3.
>
> A3:  Thanks for this comment. Please note that we performed qualitative analysis over selected samples (where the reasoning chain leads to incorrect predictions) to identify flaws in the reasoning process of LLMs, hence, low performance. We believe that different findings across various reasoning types are useful in terms of understanding mistake that is made by models in general. Moreover, this qualitative analysis supports the difficulty posed by negations and the challenges associated with longer inference rules.
>
> ---------
> R4: Use of ‘If’ as material implication in LogicBench
>
> A4: Thanks for bringing up this interesting point. While the use of ‘if’ in natural language sentences in the wild could have other interpretations such as a subjunctive, we took care in our construction so that in our dataset ‘if’ meant material implication. Specifically, our data creation pipeline ensures the logical correctness of generated context and questions where ‘if’ is used as material implication, thus maintaining the reliability of our instances.
>
> --------------
> R5: Justification of lower performance of GPT-4 on PL as compared to NM
>
> A5: Thank you for making this observation. Its explanation, as we give below, is illuminating. In the development of AI, non-monotonic logics were partly developed to formalize natural language constructs, such as “normally birds fly’’, that were not formalizable in a straightforward manner using classical mathematical logics. Thus, while it was difficult for researchers to come up with non-monotonic logics and formalize non-monotonic reasoning, the fact that they were usually motivated by natural language examples, suggests that many of the non-monotonic reasoning aspects are present in the NL text in the wild that is used in the pre-training of the ultra-large LLMs such as GPT4. On the other hand, some of the PL features are counterintuitive to humans such as if we have contradiction (a and ~a)  then everything (even unrelated) is entailed. Also, some PL features are perhaps less prevalent in human writing (on which LLMs are trained) - such as Modes Tollens. Thus your observation is not a surprise and we will add this point to the paper.
>
> ------------
> R6: Is LogicT5 pre-trained from scratch using the T5 model recipe or from a pretrained T5-Large checkpoint?
>
> A6: We train T5 on LogicBench(Aug) starting from a pre-trained T5-Large checkpoint.
>
> ---
> We continued the discussion in the next comment. Thank you!

---

> > ### Author Response · Authors · 2023-11-21
> >
> > R7: Clarification on the sentence generation process presented in Sec 3.2.1
> >
> > A7: We would like to clarify that during this step, we only generated single atomic propositions rather than the complete triplet. In the prompt, we instruct the model with entire triplets as exemplars, aiming to provide an understanding of how the generated sentences will be applied in later stages of data creation. The objective of presenting such examples is to encourage the model to produce more coherent sentences, consequently enhancing the naturalness of generated instances. While our formatting instructions explicitly indicate the generation of only atomic propositions, we provide a complete prompt in Appendix A (Figure 4) for further clarification.
> >
> > ----------
> > R8: Related work on NM Reasoning
> >
> > Thank you for this suggestion. BoardgameQA focuses on reasoning with incomplete and contradictory information, whereas, our benchmark systematically categorizes and studies different patterns for NM reasoning. We will include this in the related work in our revised version.
> >
> > -------------
> > R9: Definition of EI in Table 2
> >
> > We follow the definition of EI from https://en.wikipedia.org/wiki/Existential_instantiation
> >
> > ----
> > Note: During the rebuttal period, we conducted a human evaluation on a subset of LogicBench(Eval). We will share the results earliest possible.

---

> > > ### Comment · Reviewer_5vLY · 2023-11-21
> > > **baseline**
> > >
> > > A2: Each task is a binary classification task with a Yes/No answer. Note that, the results in Table 6 show that GPT-4 demonstrates superior performance for simpler PL inference rules such as HS and MT as compared to others in PL which is expected. However, as we introduce more complex PL inference rules (i.e., longer inference rules that are not commonly used in human writing) such as DD, BD, and CD, GPT-4 starts to show a drop in performance. This is one of the findings in the paper as discussed in Section 4.3 (“Longer inference rules are still challenging”).
> > >
> > > Can you tell us what the percentage of y/n labels were in general for these problems. There is such a divergence in accuracy between the results for Y v N, I suspect either strong class imbalance or that the prompts you used were not good (e.g. more few-shot Y examples than N).
> > >
> > > > Specifically, our data creation pipeline ensures the logical correctness of generated context and questions where ‘if’ is used > as material implication, thus maintaining the reliability of our instances.
> > >
> > > how?
> > >
> > > >On the other hand, some of the PL features are counterintuitive to humans such as if we have contradiction (a and ~a) then >everything (even unrelated) is entailed. Also, some PL features are perhaps less prevalent in human writing (on which LLMs >are trained) - such as Modes Tollens. Thus your observation is not a surprise and we will add this point to the paper.
> > >
> > >  This is a good point. Any way of bringing it out rigorously from your results?
> > >
> > > > Please note that we performed qualitative analysis over selected samples (where the reasoning chain leads to incorrect predictions)
> > >
> > > Where is this qualitative analysis?
> > >
> > > > We follow the definition of EI from https://en.wikipedia.org/wiki/Existential_instantiation
> > >
> > > How do you implement the notion of a "new constant symbol" in natural language?
> > >
> > > > During the rebuttal period, we conducted a human evaluation on a subset of LogicBench(Eval). We will share the results earliest possible
> > >
> > > this would be helpful.

---

> > > > ### Author Response · Authors · 2023-11-22
> > > >
> > > > Thanks for your quick response to our comments.
> > > >
> > > > ---
> > > > R1: Can you tell us what the percentage of y/n labels were in general for these problems?
> > > >
> > > > A1: For LogicBench(Eval), out of 1720, 531 samples are for ‘yes’ and 1189 samples are for ‘no’. LogicBench(Aug) follows the same ratio between yes and no questions. Thus, we presented a separate accuracy in Table 6, A(Yes) and A(No) to evaluate model performance. We updated this in the paper (Sec. 3.3).
> > > >
> > > > ---
> > > > R2: How do we ensure the ‘if’ as material implication in our data?
> > > >
> > > > A2: In the first stage of the data creation pipeline, we use prompts to generate propositions. In these prompts, we instruct the model with <sentences, context, question> triplets as exemplars, aiming to provide an understanding of how the generated sentences will be applied in later stages of data creation. For inference rules with ‘if’ used in them, we created prompts such that context and questions provided in those exemplars always contain the ‘if’ as material implication. Thus, this first stage ensures that generated propositions when embedded in the inference rules in the later stage yield context and questions where the ‘if’ is a material implication. Our data creation pipeline ensures the logical correctness of generated context and questions in the second stage since it uses templates developed from formal expressions of inference rules (shown in Table 4), and the ‘if’ in the inference rules that we use is the material implication ‘if’.  This further supports that ‘if’ is used as a material implication in our benchmark.
> > > >
> > > > ---
> > > > R3: This is a good point. Any way of bringing it out rigorously from your results?
> > > >
> > > > A3: Our understanding of your point “Any way of bringing it out rigorously from your results?” is that can we use our evaluations to say something about “some PL features are perhaps less prevalent in human writing”.
> > > >
> > > > Indeed, the results presented in Table 6 do give such indications. We show GPT-4 achieves ~97% accuracy (A(yes)) for simple/straightforward inference rules such as MP(FOL) and HS(PL). However, GPT-4 performance dropped to ~85% A(Yes) for MT. As the complexity of inference rules increases such as BD, CD, and DD (formal expression presented in Table 2), GPT-4 performance further deteriorates (Table 6). Thus, the evaluations support the hypothesis about the frequency of such reasoning patterns in the “training” data.
> > > >
> > > > Although not a rigorous study, we also asked GPT-4 about which type of reasoning (MP vs. MT) is found frequently on the web. Response from GPT-4 is illuminating.
> > > >
> > > > Prompt:
> > > > ```
> > > > Between the following two kinds of reasoning which is more frequently found on the web.
> > > > (1) We are given If A then B. Suppose we are told A is true. What can we say about B? Since we have A then B must be true.
> > > > (2) We are given If A then B. Suppose we are told B is false. What can we say about A?  Since we have not B then A must be false.
> > > > ```
> > > > GPT-4 Response:
> > > > ```
> > > > Both of these types of reasoning can be found on the web, but "type 1" reasoning (If A then B. Since we have A then B must be true) is more frequently encountered. This type of reasoning is often used to draw conclusions or make predictions based on the given premise. It is commonly seen in discussions, explanations, and even in everyday conversations.
> > > >
> > > > However, "type 2" reasoning (If A then B. Suppose we are told B is false. What can we say about A? Since we have not B then A must be false) is also present, but it may be less frequent. This type of reasoning, known as contrapositive reasoning, is used to deduce information about the antecedent when the consequent is known to be false. It is often used in logical arguments and discussions where the denial of the consequent leads to a conclusion about the antecedent.
> > > >
> > > > Both types of reasoning have their applications and can be found across different contexts on the web.
> > > > ```
> > > > ---
> > > > R4: Where is this qualitative analysis?
> > > >
> > > > A4: Qualitative analysis is presented in Sec. 4.3 (First section) and Appendix I where we manually evaluated the fraction of low-performing instances and validated the correctness of reasoning steps generated by LLM. Based on that, we presented findings in Sec. 4.3.
> > > >
> > > > ---
> > > > R5: Definition of EI in Table 2 (Addressing your previous point)
> > > >
> > > > A5: Thanks for highlighting this point again. This is indeed an oversight from our side. The definition of EI is indeed backward as you pointed out. What we meant was EG (​​Existential generalization - https://en.wikipedia.org/wiki/Existential_generalization). We have corrected this in the revised version.

---

> ### Author Response · Authors · 2023-11-22
>
> R6: Small-scale Human Evaluation on LogicBench(Eval)
>
> A6: Thank you for this suggestion. Please note that, in Sec. 3.3, we performed a systematic evaluation to ensure that generated instances follow the intended logical structure. To further support the integrity and reliability of the benchmark, we hired three graduate student volunteers to manually check the quality of generated data instances. Due to the limited rebuttal period, we randomly selected 5 instances from each inference rule, resulting in a total of 125 instances across 25 reasoning patterns for human evaluation. In particular, annotators are asked to provide binary answers (yes/no) for “Validity of generated context”, and “Validity of generated question” to make sure they adhere to the intended logical structure. Each instance is annotated by three different annotators.  The inter-annotator agreement, measured with raw/observed agreement is 0.956. When there was a disagreement between annotators, a majority class was preferred. We provided example task instructions at https://anonymous.4open.science/r/LogicBench-EEBB/human_eval/readme.md. Results reveal that, after majority voting, annotators provided ‘yes’ for all 125 instances showing that all given instances adhere to the intended logical structure which further supports the quality of data checked by authors. We added this study in Appendix L.

---

> > ### Comment · Reviewer_5vLY · 2023-11-22
> > **imbalance**
> >
> > >R1: Can you tell us what the percentage of y/n labels were in general for these problems?
> >
> > >A1: For LogicBench(Eval), out of 1720, 531 samples are for ‘yes’ and 1189 samples are for ‘no’. LogicBench(Aug) follows the >same ratio between yes and no questions. Thus, we presented a separate accuracy in Table 6, A(Yes) and A(No) to evaluate >model performance. We updated this in the paper (Sec. 3.3).
> >
> > Then why the severe difference in the accuracy numbrers ? I suppose you want to argue that models struggle to recognize a valid reasoning chain, but it could just as well have been poor templates in the CoT (does Flan-T5 even respond to cot? not sure). Different models might actually need different prompts, usually the right prompt can make a significant difference. It's important to know how these were chosen and how we can rule out the effect of the prompt. I see you averaged 3 prompts, but how were they chosen?
> >
> >
> >     I think what's missing the most in your paper is good analysis of the results. For example, if you had broken down the results by "length" of inference rule, you could have clearly established the conclusion you draw about its effect rather than making a claim where the evidence needs to be inferred.
> >
> > > For inference rules with ‘if’ used in them, we created prompts such that context and questions provided in those exemplars >always contain the ‘if’ as material implication
> >
> > How did you ensure this? I see this as a very hard problem since so much depends on context. e.g. 'If the butler did it, then the gardener is innocent.' would be interpreted by most speakers as either a suppositional or a subjunctive. how do you ensure that the pipeline never generates sentences like this ?

---

> ### Author Response · Authors · 2023-11-23
>
> Thank you for your response.
>
> ---
> R1: How were prompts chosen?
>
> A1: Please note that prompts presented in Appendix H.2 (used in our study) are Zero-shot-CoT prompts and we use the original paper for Zero-shot-CoT [1] as a reference to create those prompts manually.
>
> [1] https://arxiv.org/pdf/2205.11916.pdf
>
> ---
> R2: Does Flan-T5 even respond to CoT?
>
> A2: Yes, FLAN generates basic explanations for the answer when given a CoT prompt instead of generating direct predictions. Thus, we also included FLAN-T5-3B (along with GPT-3/4, ChatGPT, and Tk-instruct) in our study for comprehensive evaluation. Please see the below example response generated by FLAN-T5-3B using our prompt 3 (Appendix H.2):
> ```
> Context: If Liam finishes his work early, then he will order pizza for dinner.
> Question: If he won't order pizza for dinner, does this imply that Liam didn't finish his work early?
> FLAN (prediction): The answer is yes because if he didn't order pizza which means that he didn't finish his work early.
> ```
> From the response, we can see that FLAN generates basic explanations given the CoT prompt, not a detailed step-by-step process.
>
> ---
> R3: Analysis of results by "Length" of inference rules
>
> R3: Thank you for this suggestion. Context in the LogicBench(Eval) uses templates developed from formal expressions of inference rules, hence we use token length of context to measure the length of inference rules. The below table represents the performance of GPT-4 w.r.t. token length of context. Since inference rules from classical logic, PL and FOL, are formally expressed, we evaluate the effect of context length on the performance of GPT-4 on these inference rules.
>
> | Inference Rule  | Avg. Length | A(Yes) | A(No) |
> | ------------- | ------------- | ------------- | ------------- |
> | EG (FOL)	        | 9.4	              | 100.00%    | 100.00% |
> | MP (FOL)	        | 13.8.            | 98.41%.     | 93.75% |
> | MI (PL)	        | 16.55	      | 46.03%	   | 81.32% |
> | MT (FOL)	        | 16.55	      | 64.89%	   | 81.56% |
> | UI (FOL)	        | 16.9	      | 100.00%	   | 98.41% |
> | MT (PL)	        | 17.7	      | 85.19%	   | 94.15% |
> | HS (FOL)	        | 32.1	      | 91.49%	   | 97.93% |
> | HS (PL)	| 35.65 | 96.45% | 98.76% |
> | DS (FOL)	| 65.55	| 33.33%	| 75.32% |
> | CT (PL)	| 68.05 |	12.96% |	76.55% |
> | DS (PL)	| 68.45 | 	26.67%	| 76.22% |
> | BD (FOL)	| 93.05 |	0.00% |	76.56% |
> | CD (FOL)	| 97.75 |	33.33% |	75.46% |
> | DD (PL)	| 97.9	| 33.33%	| 75.93% |
> | BD (PL)	| 97.9	| 20.00%	| 75.03% |
> | CD (PL)	| 98.3	| 33.33%	| 76.13% |
> | DD (FOL) | 98.6	| 14.29%	| 78.30% |
>
> From the results, please note that the performance of GPT-4 decreases with the increasing token length of context. Specifically, GPT-4 often shows low performance on inference rules with longer context (e.g., CD, DD, BD, etc.).
>
> ---
> R4: How did we ensure ‘if’ as material implication in prompts?
>
> R4: Our prompts consist of <sentences, context, question> triplets as exemplars, and we (authors) created those examples manually where we ensure the examples always contain ‘if’ as material implication. Furthermore, we manually validated LogicBench(Eval) to make sure that the generated context and question adhere to the intended structure of inference rules, and the ‘if’ in the inference rules that we use is the material implication ‘if’.

---

### Author Response · Authors · 2023-11-21
**General Response**

We thank the reviewers for their insightful comments. We are encouraged that all reviewers acknowledged that our dataset can be useful in diagnosing the bottleneck of reasoning abilities of LLMs and also helpful in progressing research on advanced LLM reasoning.

We provided answers to the reviewer’s questions in the individual responses.

According to the reviewers’ comments and suggestions, we incorporated the following changes to a revised version of the paper:

- As suggested by reviewer 5vLY, we included a justification of the lower performance of GPT-4 on PL as compared to NM in Appendix M.
- As suggested by reviewer 1d9G, we updated a justification of the high few-shot performance of GPT-4 on LogicBench(Eval) in Appendix E.
- As suggested by reviewer 1d9G, we incorporated the results of the open-source LLM, LLaMa-2-7B, into Appendix G, and the results and findings of cross-type generalization in Appendix K.
- As suggested by reviewer sePi, we added a justification for the lower performance of GPT-4 on PL as compared to FOL in Appendix M.
- As suggested by reviewer Z2AS, we included clarification on using a detailed template for logical ‘or’ in Sec. 3.2.2.
- As suggested by reviewers 5vLY and Z2AS, we added a small-scale human evaluation of LogicBench(Eval) in Appendix L.
- We have incorporated all presentation-related suggestions in the revised version.

---

### Meta-Review · Area_Chair_BCM8 · 2023-12-05

**Metareview:**

The paper introduces a new benchmark dataset, LogicBench, designed to assess the logical reasoning abilities of large language models (LLMs) such as GPT-4, GPT-3, and FLAN-T5. This dataset encompasses 25 reasoning patterns across propositional, first-order, and non-monotonic logics. The reviews are mixed, acknowledging the significance of the work in advancing LLM reasoning research, particularly in underexplored areas like non-monotonic logic. However, concerns were raised about the synthetic nature of the dataset, its real-world applicability, and the surprising underperformance of LLMs in simpler logic problems, which could indicate issues with the dataset or the models themselves. While the dataset offers a comprehensive range of logic types and thorough evaluation methods, reviewers suggest that its synthetic, templated approach might not fully capture the complexity of real-world reasoning. Additionally, the paper's lack of detail in certain sections and the marginal improvements seen in model performance post-training with LogicBench were noted as drawbacks. Overall, while LogicBench is a valuable contribution to the field of LLM, it appears to have limitations in its current form, needing more detailed analysis and perhaps refinements to fully realize its potential in advancing LLM reasoning research.

**Justification For Why Not Higher Score:**

Not enough support for acceptance.

**Justification For Why Not Lower Score:**

NA

---

### Decision · Program_Chairs · 2024-01-16

Reject